# FACM: FLOW-ANCHORED CONSISTENCY MODELS

**Yansong Peng**[1,2]\*, **Kai Zhu**[1,2], **Yu Liu**[2],
**Pingyu Wu**[1,2]\*, **Hebei Li**[1], **Xiaoyan Sun**[1,3]†, **Feng Wu**[1,3]
[1]University of Science and Technology of China    [2]Tongyi Lab
[3]Institute of Artificial Intelligence, Hefei Comprehensive National Science Center
pengyansong@mail.ustc.edu.cn

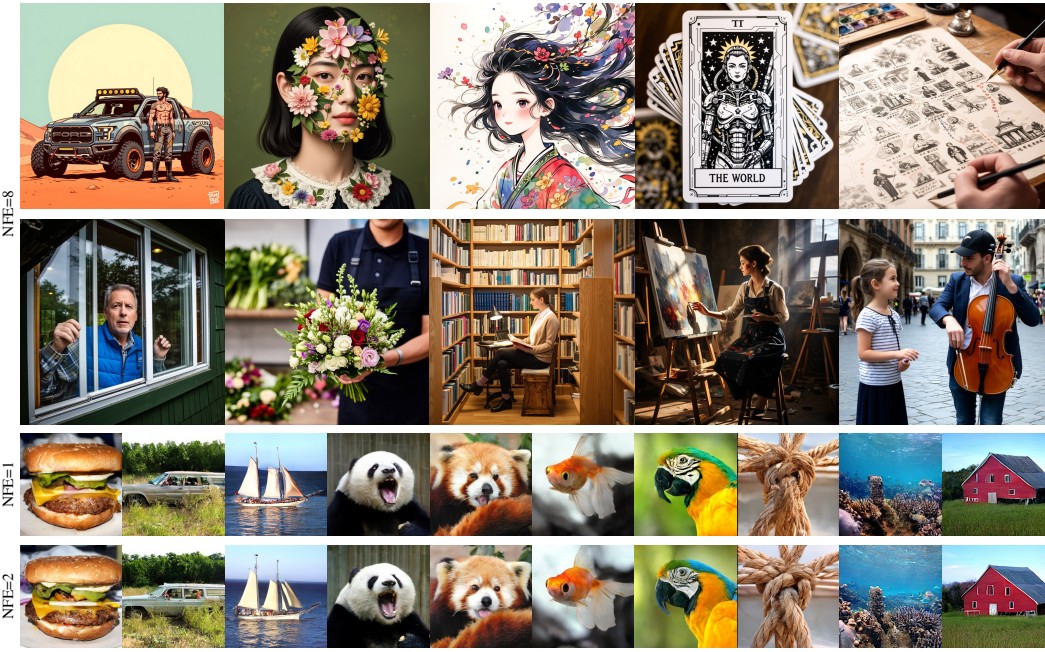

Figure 1: FACM scales effectively to high-resolution text-to-image synthesis with a 14B parameter model (top) and achieves state-of-the-art few-step generation on ImageNet $256\times256$ (bottom).

## ABSTRACT

Continuous-time Consistency Models (CMs) promise efficient few-step generation but face significant challenges with training instability. We argue this instability stems from a fundamental conflict: Training the network exclusively on a shortcut objective leads to the catastrophic forgetting of the instantaneous velocity field that defines the flow. Our solution is to explicitly anchor the model in the underlying flow, ensuring high trajectory fidelity during training. We introduce the Flow-Anchored Consistency Model (FACM), where a Flow Matching (FM) task serves as a dynamic anchor for the primary CM shortcut objective. Key to this **Flow-Anchoring** approach is a novel expanded time interval strategy that unifies optimization for a single model while decoupling the two tasks to ensure stable, architecturally-agnostic training. By distilling a pre-trained Light-ningDiT model, our method achieves a state-of-the-art FID of 1.32 with two steps (NFE=2) and 1.70 with just one step (NFE=1) on ImageNet $256\times256$. To address the challenge of scalability, we develop a memory-efficient **Chain-JVP** that resolves key incompatibilities with FSDP. This method allows us to scale FACM training on a 14B parameter model (Wan 2.2), accelerating its Text-to-Image inference from $2\times40$ to 2-8 steps. Our code and pretrained models: https://github.com/ali-vilab/FACM.

---

\*Work done during their internships at Tongyi Lab.

†Corresponding author

# 1 INTRODUCTION

As generative models scale to unprecedented sizes and applications demand real-time synthesis, the need for efficient, few-step samplers has become paramount. Consistency Models (CMs) have emerged as a promising paradigm for few-step generation (Song et al., 2023). Early successful works were largely based on discrete-time formulations (Song et al., 2023; Song & Dhariwal, 2023; Luo et al., 2023), which are inherently prone to discretization errors. While their continuous-time counterparts can circumvent these errors, they have been historically hindered by severe training instability. Recent approaches, notably sCM (Lu & Song, 2024), have made significant strides in stabilizing continuous-time training through a combination of regularization techniques and architectural modifications. Concurrently, Flow Mapping methods (Geng et al., 2025; Sabour et al., 2025; Wang et al., 2025) exemplify another line of research that has aimed to stabilize training. By reformulating the shortcut objective itself, these methods either model the "average velocity" to arbitrary endpoints, or introduce additional self-consistency constraints between multi-timesteps. Although these methods provide stable few-step sampling, they fail to address the root cause of instability. Their reliance on a single, over-coupled objective to learn the flow and shortcut simultaneously prevents explicit task decoupling and compromises perfect trajectory fidelity.

This paper addresses the root cause of instability in the continuous CM objective from a novel perspective. We posit that the standard continuous CM objective, while powerful for learning a direct "shortcut" across a probability flow, is inherently unstable when trained in isolation. This is because the approach implicitly assumes the model has a robust understanding of the underlying flow. However, training exclusively on the shortcut objective can induce catastrophic forgetting of this flow, leading to training collapse. Our key insight is that stability can be achieved by explicitly anchoring the model in the very flow it is shortcutting.

The most direct way to achieve this **Flow-Anchoring** is to re-introduce the explicit training of the **instantaneous velocity field** that defines the flow. We propose that an objective based on Flow Matching (FM) (Lipman et al., 2022) can act as a crucial anchor, enabling the primary shortcut objective to be trained effectively. Based on this principle, we introduce the Flow-Anchored Consistency Model (FACM), which employs a simple yet effective training strategy combining two distinct objectives:

- **Flow-Anchoring Objective** that learns the flow's velocity field to provide stability.
- **Shortcut Objective** that learns the efficient one-step consistency mapping.

Our architecturally-agnostic method is stabilized by an innovative **expanded time interval** strategy that decouples these objectives into distinct domains, while forming a continuous target that unifies the optimization for a single model, supporting high-fidelity and stable training. By distilling a pre-trained LightningDiT model, our approach sets new state-of-the-art FID scores of 1.70 (NFE=1) and 1.32 (NFE=2) on the ImageNet 256×256 benchmark. To enable scalability, we solve a key memory bottleneck caused by the Jacobian-Vector Product (JVP), which is incompatible with modern training techniques like Fully Sharded Data Parallel (FSDP). We introduce a memory-efficient Chain-JVP that computes derivatives sequentially by module, avoiding prohibitive memory spikes. This allows us to train a 14B parameter model and accelerate its inference from 2×40 to just 2-8 steps.

# 2 BACKGROUND

**Diffusion and Flow Matching.** Generative models aim to transform a prior distribution $p_0$ (e.g., $\mathcal{N}(0, I)$) to a data distribution $p_1$. A dominant approach is Diffusion Models (Ho et al., 2020; Song et al., 2020; Karras et al., 2022), which learn to reverse a predefined noising process. Flow Matching (FM) (Lipman et al., 2022; Liu et al., 2022; Albergo & Vanden-Eijnden, 2022; Albergo et al., 2023; Kingma & Gao, 2024) offers a more direct framework to learn the probability flow ODE by regressing its output against a target velocity $\mathrm{d}\boldsymbol{x}_t/\mathrm{d}t = \boldsymbol{v}_\theta(\boldsymbol{x}_t, t)$. A common approach uses the OT-FM path $\boldsymbol{x}_t = (1-t)\boldsymbol{x}_0 + t\boldsymbol{x}_1$ between a noise sample $\boldsymbol{x}_0$ and a data sample $\boldsymbol{x}_1$, which has a constant conditional velocity of $\boldsymbol{x}_1 - \boldsymbol{x}_0$. This leads to the practical FM objective:

$$\mathcal{L}_{\mathrm{FM}}(\theta) = \mathbb{E}_{t, \boldsymbol{x}_0, \boldsymbol{x}_1} \|\boldsymbol{v}_\theta(\boldsymbol{x}_t, t) - (\boldsymbol{x}_1 - \boldsymbol{x}_0)\|_2^2. \quad (1)$$

**Consistency Models.** Consistency Models (CMs) (Song et al., 2023) are trained to map any point $\boldsymbol{x}_t$ on an ODE trajectory directly to its endpoint $\boldsymbol{x}_1$ in a single evaluation. While early successful

works were largely based on discrete-time formulations that are prone to discretization errors (Song & Dhariwal, 2023; Geng et al., 2024; Luo et al., 2023; Zheng et al., 2024), our work focuses on the continuous-time formulation. This approach requires the total derivative of the model's output to be zero: $\frac{\mathrm{d}f_\theta(\boldsymbol{x}_t,t)}{\mathrm{d}t} = 0$. With the standard parameterization $f_\theta(\boldsymbol{x}_t,t) = \boldsymbol{x}_t + (1-t)\boldsymbol{F}_\theta(\boldsymbol{x}_t,t)$ and the boundary condition $f_\theta(\boldsymbol{x}_1,1) = \boldsymbol{x}_1$, this implies the network $\boldsymbol{F}_\theta$ must satisfy:

$$\boldsymbol{F}_\theta(\boldsymbol{x}_t,t) = \boldsymbol{v} + (1-t)\frac{\mathrm{d}\boldsymbol{F}_\theta(\boldsymbol{x}_t,t)}{\mathrm{d}t}. \tag{2}$$

Here, $\boldsymbol{v}$ represents the conditional velocity $\boldsymbol{x}_1 - \boldsymbol{x}_0$ from the underlying flow. In the distillation paradigm, this velocity is provided by a pre-trained FM teacher. This objective, relying on a Jacobian-vector product (JVP) for the derivative term, is notoriously unstable to train (Lu & Song, 2024). Recently, Flow Mapping methods (Zhou et al., 2025; Geng et al., 2025; Sabour et al., 2025; Wang et al., 2025; Guo et al., 2025) have extended consistency models with a unified objective, but they do not address the root cause of instability and compromise perfect trajectory fidelity.

# 3 FLOW-ANCHORED CONSISTENCY MODELS (FACM)

This section first analyzes the core instability of continuous-time Consistency Models (CMs), identifying the "missing anchor" as the root cause. We then present our solution, the Flow-Anchored Consistency Model (FACM), detailing its mixed-objective training strategy. Our analysis reframes the challenge of training continuous-time Consistency Models. We argue that the instability is not an inherent flaw of the shortcut objective itself, but a consequence of training on it in isolation, which causes the model to lose its anchor in the flow's underlying velocity field.

## 3.1 REVISIT THE SHORTCUT TARGET OF CONSISTENCY MODELS

To understand the mechanics of the generative shortcut, we first re-examine the consistency model's learning objective. The goal of a consistency function $f_\theta(\boldsymbol{x}_t,t)$ is to map any point $\boldsymbol{x}_t$ on an ODE trajectory to its endpoint $\boldsymbol{x}_1$. Using the OT-FM parameterization $f_\theta(\boldsymbol{x}_t,t) = \boldsymbol{x}_t + (1-t)\boldsymbol{F}_\theta(\boldsymbol{x}_t,t)$, the ideal shortcut $f_\theta(\boldsymbol{x}_t,t) = \boldsymbol{x}_1$ can only be achieved if the network $\boldsymbol{F}_\theta$ learns to predict a very specific quantity:

$$\boldsymbol{x}_t + (1-t)\boldsymbol{F}_\theta(\boldsymbol{x}_t,t) = \boldsymbol{x}_1 \quad \Rightarrow \quad \boldsymbol{F}_\theta(\boldsymbol{x}_t,t) = \frac{\boldsymbol{x}_1 - \boldsymbol{x}_t}{1-t}. \tag{3}$$

This term has a clear physical interpretation: it is the average velocity required to travel from point $\boldsymbol{x}_t$ to the endpoint $\boldsymbol{x}_1$ in the remaining time $1-t$. We denote this quantity as $\overline{\boldsymbol{v}}(\boldsymbol{x}_t,t)$. Thus, the task of learning the one-step shortcut is equivalent to training $\boldsymbol{F}_\theta$ to predict this average velocity.

Now, we investigate the properties that this average velocity field must satisfy. From its definition in Eq. 3, we have $(1-t)\overline{\boldsymbol{v}}(\boldsymbol{x}_t,t) = \boldsymbol{x}_1 - \boldsymbol{x}_t$. Differentiating both sides with respect to $t$ using the product rule gives:

$$\frac{d}{dt}\left((1-t)\cdot\overline{\boldsymbol{v}}(\boldsymbol{x}_t,t)\right) = -\frac{\mathrm{d}\boldsymbol{x}_t}{\mathrm{d}t} \quad \Rightarrow \quad -\overline{\boldsymbol{v}}(\boldsymbol{x}_t,t) + (1-t)\frac{\mathrm{d}\overline{\boldsymbol{v}}(\boldsymbol{x}_t,t)}{\mathrm{d}t} = -\boldsymbol{v}(\boldsymbol{x}_t,t). \tag{4}$$

Rearranging the terms, we arrive at a key differential identity that the true average velocity field must satisfy:

$$\overline{\boldsymbol{v}}(\boldsymbol{x}_t,t) = \boldsymbol{v}(\boldsymbol{x}_t,t) + (1-t)\frac{\mathrm{d}\overline{\boldsymbol{v}}(\boldsymbol{x}_t,t)}{\mathrm{d}t}. \tag{5}$$

This identity is formally identical to the continuous-time CM learning objective (Eq. 2) and the Meanflow identity ($r \equiv 1$). This confirms that the CM objective directly forces the network $\boldsymbol{F}_\theta$ to learn the properties of an average velocity field, thus enabling the one-step generation shortcut.

## 3.2 THE SOURCE OF INSTABILITY: LOSING THE FLOW ANCHOR

While Eq. 2 correctly identifies the target, its practical implementation via the training objective $T = \boldsymbol{v} + (1-t)\frac{\mathrm{d}\boldsymbol{F}_{\theta-}(\boldsymbol{x}_t,t)}{\mathrm{d}t}$ is notoriously unstable. The core of this instability lies in the target's self-referential nature. This dependency creates two fundamental, intertwined problems:

**Missing Instantaneous Velocity Field Supervision.** The target $T$ explicitly depends on the instantaneous velocity $v$. However, the CM objective only enforces a loss on the final prediction $F_\theta$. There is no explicit mechanism to ensure that the model's learned dynamics remain faithful to the underlying instantaneous velocity field $v(\mathbf{x}_t, t)$. The model is being asked to learn the integral of a function (average velocity) without being explicitly taught the function itself (instantaneous velocity).

**Self-Referential Derivative Estimation.** This lack of direct supervision on $v$ makes the derivative term, $\frac{\mathrm{d}F_{\theta^-}}{\mathrm{d}t}$, highly unstable. The total derivative, expanded via the chain rule, is:

$$\frac{\mathrm{d}F_{\theta^-}(\boldsymbol{x}_t, t)}{\mathrm{d}t} = (\nabla_{\boldsymbol{x}_t} F_{\theta^-})\boldsymbol{v} + \frac{\partial F_{\theta^-}}{\partial t}. \tag{6}$$

The network is optimized to estimate its own derivative to satisfy the consistency identity in Eq. 2. Ideally, this process should facilitate a smooth transition, evolving the model from predicting the instantaneous velocity field to an average velocity field that satisfies this identity. However, without a stable anchor in the underlying flow, the model's output $F_\theta$ quickly begins to drift. This drift has a critical consequence: the derivative term in the identity grows to dominate the ground-truth velocity $v$, effectively diluting its supervisory signal. At this point, satisfying the identity no longer converges to the boundary condition. The training target thus becomes noisy and erratic, creating a vicious cycle that rapidly amplifies errors and ultimately leads to training collapse.

These two issues stem from the same fundamental problem: the CM objective is ungrounded. It lacks a stable foundation in the very flow it is supposed to shortcut. The antidote is to re-introduce the explicit supervision of the instantaneous velocity field $v$ via a Flow Matching objective. This provides a stable **anchor** for the model's internal dynamics, ensuring that the model's gradient field is well-behaved, which directly stabilizes the derivative term in the CM objective and allows the primary shortcut objective to be learned effectively. We term this principle **Flow-Anchoring**.

## 3.3 THE FACM TRAINING STRATEGY

Based on our analysis, we introduce the Flow-Anchored Consistency Model (FACM). Instead of requiring specialized architectures, FACM employs a simple and effective training strategy that mixes two complementary objectives: one for stability (the anchor) and one for efficiency (the accelerator).

### 3.3.1 THE FACM OBJECTIVE: AN ANCHOR AND AN ACCELERATOR

The FACM training approach harnesses the stability of **Flow-Anchoring** (the FM task) and the efficiency of direct shortcut learning (the CM task) within a single training loop. The overall training loss, $\mathcal{L}_{\text{FACM}}$, is a sum of two complementary objectives:

$$\mathcal{L}_{\text{FACM}} = \mathcal{L}_{\text{FM}} + \mathcal{L}_{\text{CM}} \tag{7}$$

To enable the model to distinguish between the two tasks, each objective uses a distinct conditioning signal, $c_{\text{FM}}$ and $c_{\text{CM}}$, which we detail in Section 3.3.2.

**Flow Matching (FM) Loss (The Anchor).** This loss component anchors the model by regressing its output towards the instantaneous velocity $v$. The target $v$ is constructed with a base velocity $v_{\text{base}}$ and an optional classifier-free guidance (CFG) (Ho & Salimans, 2022) term:

$$\boldsymbol{v} = \boldsymbol{v}_{\text{base}} + w \cdot (\boldsymbol{v}_{\text{cond}} - \boldsymbol{v}_{\text{uncond}}), \tag{8}$$

where $w$ is the guidance scale. The definitions of these components vary by training paradigm. For from-scratch training, the base is the conditional velocity, $v_{\text{base}} = \boldsymbol{x}_1 - \boldsymbol{x}_0$, and the guidance term is derived from the online model $F_\theta$ itself. In distillation, the model is initialized with weights from a pre-trained FM model. A non-trainable copy of these weights, denoted as the "teacher" $F_\delta$, provides all velocity components for the target, with $v_{\text{base}} = v_{\text{uncond}} = F_\delta(\boldsymbol{x}_t, \emptyset)$, making the formula equivalent to standard CFG. Without CFG ($w = 1$), the target simply defaults to $v_{\text{cond}}$. The FM loss then combines an L2 term with a cosine similarity term $L_{\cos}(\boldsymbol{a}, \boldsymbol{b}) = 1 - (\boldsymbol{a} \cdot \boldsymbol{b})/(\|\boldsymbol{a}\|_2\|\boldsymbol{b}\|_2)$:

$$\mathcal{L}_{\text{FM}}(\theta) = \mathbb{E}\left[\|F_\theta(\boldsymbol{x}_t, c_{\text{FM}}) - \boldsymbol{v}\|_2^2 + L_{\cos}(F_\theta(\boldsymbol{x}_t, c_{\text{FM}}), \boldsymbol{v})\right]. \tag{9}$$

**Consistency Model (CM) Loss (The Accelerator).** This component acts as an accelerator, training the model to learn the generative shortcut. We interpret the consistency condition (Eq. 2) as a

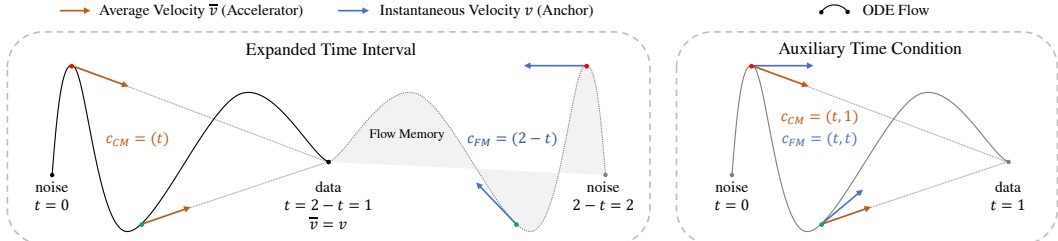

Figure 2: Two implementation strategies for the mixed-objective function in FACM. (A) **Expanded Time Interval** (default): The time domain is conceptually doubled, showing the same ODE flow on two intervals. The CM task is performed on $t \in [0, 1]$. To perform the FM task at a point $t$ on the flow, the model is conditioned on $c_{\text{FM}} = 2 - t$, which maps the time to the alternate interval $[1, 2]$ to distinguish the two tasks. (B) **Auxiliary Time Condition**: An additional time condition $r$ is introduced to the model. When $r = 1$, the model learns the CM task (average velocity from $t$ to 1, orange); when $r = t$, it learns the FM task (instantaneous velocity at t, blue).

fixed-point problem, $\boldsymbol{F}_\theta = \mathcal{T}(\boldsymbol{F}_\theta)$, where the operator is $\mathcal{T}(\boldsymbol{F}) \triangleq \boldsymbol{v} + (1 - t)\frac{\mathrm{d}\boldsymbol{F}}{\mathrm{d}t}$. The training objective is designed to solve this problem stably and iteratively. First, we compute the consistency residual $\boldsymbol{g}$ of the stop-gradient model $\boldsymbol{F}_{\theta^-}$ ($\boldsymbol{F}_{\theta^-} = \text{sg}(\boldsymbol{F}_\theta)$) :

$$\boldsymbol{g} = \boldsymbol{F}_{\theta^-}(\boldsymbol{x}_t, c_{\text{CM}}) - \mathcal{T}(\boldsymbol{F}_{\theta^-}) = \boldsymbol{F}_{\theta^-}(\boldsymbol{x}_t, c_{\text{CM}}) - \left( \boldsymbol{v} + (1 - t)\frac{\mathrm{d}\boldsymbol{F}_{\theta^-}(\boldsymbol{x}_t, c_{\text{CM}})}{\mathrm{d}t} \right). \tag{10}$$

This residual $\boldsymbol{g}$ is then clamped to the range $[-1, 1]$ to prevent extreme gradients. A perturbed target is then formed as:

$$\boldsymbol{v}_{\text{tar}} = \boldsymbol{F}_{\theta^-}(\boldsymbol{x}_t, c_{\text{CM}}) - \alpha(t) \cdot \boldsymbol{g}. \tag{11}$$

Substituting the definition of $\boldsymbol{g}$ reveals the target's structure as a relaxation step for the fixed-point iteration:

$$\boldsymbol{v}_{\text{tar}} = (1 - \alpha(t))\boldsymbol{F}_{\theta^-} + \alpha(t)\mathcal{T}(\boldsymbol{F}_{\theta^-}). \tag{12}$$

This formulation provides a stable, interpolated learning target between the current model's output and the ideal consistency target. The final CM loss component uses a norm L2 loss, $L_{\text{norm}}$, and is modulated by weighting functions $\alpha(t)$ and $\beta(t)$ (detailed in Appendix A.3 and A.4(c)):

$$\mathcal{L}_{\text{CM}}(\theta) = \mathbb{E}\left[ \beta(t) \cdot L_{\text{norm}}(\boldsymbol{F}_\theta(\boldsymbol{x}_t, c_{\text{CM}}), \boldsymbol{v}_{\text{tar}}) \right]. \tag{13}$$

The combination of the interpolated target $\boldsymbol{v}_{\text{tar}}$ from the CM loss and the stabilizing flow anchor from the FM loss enables effective training. It is important to note that our specific choices for weighting and loss functions are designed to accelerate convergence, not as prerequisites for stability, which is already guaranteed by the Flow-Anchoring principle.

### 3.3.2 IMPLEMENTATION OF THE MIXED OBJECTIVE

A key design question is how to encode the distinct conditioning signals, $c_{\text{FM}}$ for the FM loss and $c_{\text{CM}}$ for the CM loss, that tell the model which velocity to predict. While this conditioning can include various information like class labels, for clarity in this section, we focus only on the time-based components. We explore an effective strategy for this (Figure 2):

**Expanded Time Interval.** We innovatively propose leveraging an expanded time domain to distinguish between the two tasks, a strategy that requires no architectural modifications. The primary CM task operates on the interval $t \in [0, 1]$, using the time directly as the condition: $c_{\text{CM}} = t$. To perform the FM task at the same point $\boldsymbol{x}_t$ (defined by $t$), we signal this by mapping $t$ to the alternate interval $[1, 2]$. This is done by setting the conditioning input to $c_{\text{FM}} = 2 - t$, which makes the two conditions decoupled, symmetric, and easily distinguishable. This mapping also ensures continuity at the boundary $t = 1$, as the CM learning objective from Eq. 2 naturally converges to the FM objective's target at the boundary:

$$\lim_{t \to 1^-} \left( \boldsymbol{v} + (1 - t)\frac{\mathrm{d}\boldsymbol{F}_\theta(\boldsymbol{x}_t, t)}{\mathrm{d}t} \right) = \boldsymbol{v}. \tag{14}$$

This ensures a smooth transition between the two learning regimes.

---

**Algorithm 1** FACM Training

---

**Require:** Online model $\boldsymbol{F}_\theta$, pretrained teacher $F_\delta$, metrics $\mathcal{L}_{\text{FM}}, \mathcal{L}_{\text{CM}}$

 1: Sample $\boldsymbol{x}_0, \boldsymbol{x}_1, t$
 2: Define $c_{\text{CM}}, c_{\text{FM}}$ based on $t$ (see Sec 3.3.2)
 3: $\boldsymbol{x}_t \leftarrow (1 - t)\boldsymbol{x}_0 + t\boldsymbol{x}_1$
 4: $\boldsymbol{v} \leftarrow F_\delta(\boldsymbol{x}_t, c_{\text{FM}})$                  ▷ For training from scratch, use $\boldsymbol{x}_1 - \boldsymbol{x}_0$ instead
 5: $F_{\text{FM}} \leftarrow \boldsymbol{F}_\theta(\boldsymbol{x}_t, c_{\text{FM}})$
 6: $F_{\text{CM}}, \nabla_t \boldsymbol{F}_\theta \leftarrow \text{JVP}(\boldsymbol{F}_\theta, (\boldsymbol{x}_t, c_{\text{CM}}), (\boldsymbol{v}, 1))$         ▷ Simultaneous forward pass and JVP
 7: $\overline{\boldsymbol{v}} \leftarrow \boldsymbol{v} + (1 - t) \cdot \text{sg}(\nabla_t \boldsymbol{F}_\theta)$
 8: $\boldsymbol{v}_{\text{tar}} \leftarrow (1 - \alpha(t)) \cdot \text{sg}(F_{\text{CM}}) + \alpha(t) \cdot \overline{\boldsymbol{v}}$            ▷ Compute relaxation target
 9: $\mathcal{L}_{\text{Total}} \leftarrow \mathcal{L}_{\text{FM}}(F_{\text{FM}}, \boldsymbol{v}) + \mathcal{L}_{\text{CM}}(F_{\text{CM}}, \boldsymbol{v}_{\text{tar}})$

---

**Auxiliary Condition with a Second Timestamp.** Alternatively, another intuitive approach is to introduce a second time variable, $r$, to the model, making its full conditioning a tuple of $(t, r)$. We then define $c_{\text{CM}} = (t, 1)$ and $c_{\text{FM}} = (t, t)$. This means the model signature is effectively $\boldsymbol{F}_\theta(\boldsymbol{x}_t, t, r)$. When $r = 1$, the model is trained on the CM task (predicting average velocity from $t$ to 1). When $r = t$, the model is trained on the FM task (predicting instantaneous velocity at $t$, or from $t$ to $t$). We can provide this auxiliary condition $r$ to the model through a zero-initialized time embedder, which does not alter its original structure or initial output.

As shown in our ablations (Table 3), while both methods effectively stabilize training, the **Expanded Time Interval** strategy consistently yields the best performance. We attribute this to its use of highly distinct time domains ($[0, 1]$ vs. $[1, 2]$), which provide clearer, more separable conditioning signals for the two tasks compared to the subtler differences in the Auxiliary Time Condition (e.g., $(t, 1)$ vs. $(t, t)$). For clarity, if $t = 0$ represents the data distribution (i.e., $\boldsymbol{x}_t = t\boldsymbol{x}_0 + (1 - t)\boldsymbol{x}_1$), the conditions for the two strategies would be $t$ vs. $-t$ and $(t, 0)$ vs. $(t, t)$, respectively.

### 3.3.3 TRAINING ALGORITHM AND SCALABLE CHAIN-JVP IMPLEMENTATION

With the objective functions and conditioning signals defined, we present the complete FACM training strategy in Algorithm 1. A key component of this algorithm is the computation of the total derivative $\nabla_t \boldsymbol{F}_\theta$ in the CM loss (Line 7), performed using a Jacobian-vector product (JVP).

The JVP computation, however, presents critical bottlenecks when using modern acceleration techniques. While its incompatibility with components like Flash Attention 2 (Dao, 2024) can be resolved using methods from sCM (Lu & Song, 2024), a more fundamental memory bottleneck emerges from its conflict with Fully Sharded Data Parallel (FSDP) (Zhao et al., 2023). Standard JVP implementations require the model's full parameters $\theta$ to be materialized on the device, forcing an `all_gather` operation in an FSDP setup. This reconstructs the entire parameter set on each GPU, causing a prohibitive memory spike that makes training models with over ten billion (10B) parameters impossible. To overcome this, we leverage the chain rule. For a network composed of modules $\boldsymbol{F}_\theta = f_L \circ \cdots \circ f_1$, the JVP can be computed sequentially:

$$J_{\boldsymbol{F}_\theta}(\boldsymbol{z}) \cdot \boldsymbol{v} = J_{f_L}(\boldsymbol{z}_{L-1}) \cdot (\cdots \cdot (J_{f_2}(\boldsymbol{z}_1) \cdot (J_{f_1}(\boldsymbol{z}_0) \cdot \boldsymbol{v})) \dots) \tag{15}$$

where $\boldsymbol{z}_i = f_i(\boldsymbol{z}_{i-1})$ is the intermediate output. Our approach computes the JVP for each module sequentially, embedding this operation within the FSDP logic. Its speed is consistent with a standalone JVP pass, adding only standard FSDP overhead. This ensures that only one module's parameters are materialized at a time. Consequently, peak memory depends on the largest module, not the entire model, and the resulting memory savings grow with the model's parameter count.

In summary, the principle of **Flow-Anchoring** offers a robust and fundamental solution. While other methods achieve stability, they do so with certain limitations. For instance, sCM (Lu & Song, 2024) requires architectural modifications to its normalization layers, limiting its adaptability to large, pre-trained models. Other approaches like MeanFlow (Geng et al., 2025), while clever, present a trade-off: by treating the instantaneous velocity as merely an edge case ($r = t$) of the primary average velocity objective, the learning tasks become over-coupled. As a result, the supervisory

Table 1: **Few-step generation on CIFAR-10 and ImageNet 256×256**. "×2" indicates that CFG doubles the NFE per step. Our method sets a new state-of-the-art on both datasets.

**Unconditional CIFAR-10**

| Method | NFE | FID ($\downarrow$) |
|---|---|---|
| *Multi-NFE Baselines* | | |
| DPM-Solver++ (Lu et al., 2022) | 10 | 2.91 |
| EDM (Karras et al., 2022) | 35 | 2.01 |
| *Few-NFE Methods (NFE=1)* | | |
| iCT (Song & Dhariwal, 2023) | 1 | 2.83 |
| eCT (Geng et al., 2024) | 1 | 3.60 |
| sCM (sCT) (Lu & Song, 2024) | 1 | 2.85 |
| IMM (Zhou et al., 2025) | 1 | 3.20 |
| MeanFlow (Geng et al., 2025) | 1 | 2.92 |
| **FACM (Ours)** | 1 | **2.69** |
| *Few-NFE Methods (NFE=2)* | | |
| TRACT (Berthelot et al., 2023) | 2 | 3.32 |
| CD (LPIPS) (Song et al., 2023) | 2 | 2.93 |
| iCT-deep (Song & Dhariwal, 2023) | 2 | 2.24 |
| ECT (Geng et al., 2024) | 2 | 2.11 |
| sCM (sCT) (Lu & Song, 2024) | 2 | 2.06 |
| IMM (Zhou et al., 2025) | 2 | 1.98 |
| **FACM (Ours)** | 2 | **1.87** |

**Class-Conditional ImageNet 256×256**

| Method | Params | NFE | FID ($\downarrow$) |
|---|---|---|---|
| *Multi-NFE Baselines* | | | |
| SiT-XL/2 (Ma et al., 2024) | 675M | 250×2 | 2.06 |
| DiT-XL/2 (Peebles & Xie, 2023) | 675M | 250×2 | 2.27 |
| REPA (Yu et al., 2025) | 675M | 250×2 | 1.42 |
| LightningDiT (Yao et al., 2025) | 675M | 250×2 | 1.35 |
| *Few-NFE Methods (NFE=1)* | | | |
| iCT (Song & Dhariwal, 2023) | 675M | 1 | 34.24 |
| Shortcut (Frans et al., 2025) | 675M | 1 | 10.60 |
| MeanFlow (Geng et al., 2025) | 676M | 1 | 3.43 |
| **FACM (Ours)** | 675M | 1 | **1.70** |
| *Few-NFE Methods (NFE=2)* | | | |
| iCT (Song & Dhariwal, 2023) | 675M | 2 | 20.30 |
| IMM (Zhou et al., 2025) | 675M | 1×2 | 7.77 |
| MeanFlow (Geng et al., 2025) | 676M | 2 | 2.20 |
| **FACM (Ours)** | 675M | 2 | **1.32** |

signal for the underlying flow is often diluted, which we have observed can lead to training collapses and underfitting. In contrast, FACM provides a more direct and principled solution. Through our innovative expanded time interval strategy, the anchoring and shortcut tasks are functionally decoupled into distinct domains. This ensures the flow anchor receives a clear, undiluted supervisory signal at all times, forcing the model to maintain a stable and high-fidelity representation of the flow. This robust theoretical foundation, combined with our scalable **Chain-JVP** implementation, makes FACM not only stable but also highly practical for training models at an unprecedented scale.

# 4 EXPERIMENTS

## 4.1 EXPERIMENTAL SETUP

We empirically validate FACM on image generation benchmarks, including CIFAR-10 (Krizhevsky & Hinton, 2009) and ImageNet 256×256 (Deng et al., 2009). We evaluate models based on Fréchet Inception Distance (FID) (Heusel et al., 2017) and the Number of Function Evaluations (NFE). FACM can be trained from scratch or by distilling a pre-trained model. Our default experimental setup involves a two-stage process. We first pre-train a FM model, incorporating our mixed-objective conditioning as detailed in Appendix A.4 **(a)** to accelerate the subsequent distillation. We then distill this teacher using the FACM strategy. For few-step inference, we follow the standard multi-step sampling procedure for CMs as described in Appendix A.4 **(b)**. Further details on our experimental settings are provided in Appendix A.4. To demonstrate scalability, we also distill a 14B parameter model (Wan 2.2) on the text-to-image (T2I) task, achieving high-fidelity generation in just 2-8 steps.

## 4.2 MAIN RESULTS

### 4.2.1 COMPARISON WITH STATE-OF-THE-ART

As shown in Table 1, FACM achieves state-of-the-art results on both CIFAR-10 and ImageNet 256×256. Specifically, our method achieves FIDs of 1.70 (NFE=1) and 1.32 (NFE=2) on ImageNet 256×256 by training a LightningDiT model in latent-space, and 2.69 (NFE=1) and 1.87 (NFE=2) on CIFAR-10 by training a DDPM++ model (Ho et al., 2020) in pixel-space, significantly outperforming previous methods on both benchmarks. Remarkably, our few-step model even surpasses some multi-step baselines that require hundreds of function evaluations.

## 4.3 ABLATION STUDY ON THE TRAINING STRATEGY

We conduct ablation studies to validate our claims regarding the training strategy. We test on the ImageNet 256×256 dataset by distilling a pre-trained LightningDiT model. The results provide

Table 2: FID scores (NFE=2) on ImageNet 256×256 for different few-step methods applied to various backbone architectures. † indicates our reproduction.

| Backbone | Baseline (NFE=250×2) | sCM† | MeanFlow† | FACM (Ours) |
|---|---|---|---|---|
| SiT-XL/2 | 2.06 | 2.83 | 2.27 | **2.07** |
| REPA | 1.42 | 2.25 | 1.88 | **1.52** |
| DiT-XL/2 | 2.27 | 2.91 | 2.62 | **2.31** |
| LightningDiT | 1.35 | 1.94 | 1.74 | **1.32** |

Table 3: Ablation on stabilization strategies. All methods are distilled from the same LightningDiT teacher. †: Our reproduction. ∗: For sCM, more epochs yield worse results.

| Method | Params | FM epochs | CM epochs | FID (NFE=1, ↓) | Stable |
|---|---|---|---|---|---|
| sCM (w/o pixel norm.) | 675M | 800 | - | - | × |
| sCM (w/ pixel norm.)† | 676M | 600 | 30* | 3.04 | ✓ |
| MeanFlow † | 676M | 800 | 200 | 2.75 | ✓ |
| FACM (Auxiliary Condition) | 676M | 800 | 200 | 1.97 | ✓ |
| FACM (Expanded Interval) | 675M | 800 | 200 | **1.81** | ✓ |
| **Training from scratch methods** | | | | | |
| MeanFlow † | 676M | 0 | 1120 | 2.65 | ✓ |
| FACM (Expanded Interval) | 675M | 0 | 800 | 2.27 | ✓ |

strong evidence for our central claim: the presence of the FM objective is the critical stabilizing anchor.

**Different Architectures.** To demonstrate the architectural agnosticism of our approach, we apply FACM, sCM, and MeanFlow to a range of state-of-the-art architectures, including SiT-XL/2, REPA, DiT-XL/2, and LightningDiT. All methods are distilled from their respective multi-step FM models. As shown in Table 2, FACM consistently achieves the lowest FID scores across all tested backbones. This highlights that Flow-Anchoring is a fundamental principle for stabilizing consistency training that is not limited to a specific model design.

**Stabilization Strategy.** To ensure a fair comparison, we distill sCM, MeanFlow, and FACM from an identical LightningDiT teacher (reproduction details in Appendix A.4 **(c)**). As shown in Table 3, FACM achieves superior results due to its principled approach to stability without requiring architectural changes. In contrast, sCM's stability is limited, depending on architectural modifications (pixel normalization) and sensitive hyperparameter tuning. MeanFlow achieves robustness but at the cost of an over-coupled objective ($u(z, t, t) = v(z, t)$) that hinders optimization by diluting the essential path modeling task. FACM's explicit task separation proves more effective, as it allows the model to stably learn the shortcut while remaining anchored to the teacher's flow.

**Sensitivity to Teacher Model Quality.** As shown in Figure 4(a), FACM's performance monotonically improves with teacher quality. This demonstrates that by explicitly anchoring the teacher's complex flow, our method can consistently benefit from stronger teachers. This confirms FACM acts as a high-fidelity trajectory compression rather than a lossy compromise on the pre-trained flow.

**Ablation on Key Components.** As shown in Table 4, introducing Flow-Anchoring with our *Expanded Time Interval* decouples the FM and CM tasks, yielding faster convergence. Fidelity is further improved as shortcut interpolation ($\alpha$) and beta weighting ($\beta$) ensure a smooth transition to FM supervision as $t \to 1$, while residual clamping suppresses gradient spikes. Together, these components stably guide the learning dynamic via the FM anchor, leading to significantly better trajectory fidelity.

**Sensitivity to FM Loss Weight.** The FM loss is a prerequisite for stability, but the minimum required weight $\lambda_{FM}$ depends on the model's initialization. Our investigation reveals a nuanced picture that strongly supports a simple default choice (e.g., $\lambda_{FM} = 1.0$). As summarized in Table 5 (left vs. right), the non-finetuned setting requires at least $\lambda_{FM} \geq 0.1$ to avoid collapse, whereas the finetuned setting remains stable with $\lambda_{FM}$ as low as $10^{-8}$. These results lead to a key conclusion: while a non-zero FM weight is essential, FACM is highly robust to the specific weight across several orders of magnitude once stability is achieved. This robustness, which stems from our decoupled design, makes a direct summation a simple, effective, and reliable choice that avoids costly hyperparameter tuning.

Table 4: Ablation study on key techniques on ImageNet 256×256.

| Configuration | FID@Epochs 10 (NFE=1, ↓) | Collapse |
|---|---|---|
| MeanFlow / Fixed $r = 1$ MeanFlow (0% FM) | 372.3-391.5 | Yes |
| MeanFlow (75% FM) | 43.03 | No |
| Fixed $r = 1$ MeanFlow (75% FM) | 15.54 | No |
| **w/ Flow Anchoring (Expanded Time Interval)** | **4.31** | **No** |
| w/ Interpolation ($\alpha(t) = 1$) | 3.42 | No |
| w/ Residual Clamping | 2.86 | No |
| **w/ Beta Weighting ($\beta(t) = 1$) (FACM)** | **2.51** | **No** |

Table 5: Sensitivity to $\lambda_{\text{FM}}$ under two settings. Left: model **not** pre-finetuned on $1 < t < 2$. Right: model **is** pre-finetuned on $1 < t < 2$.

| FM Loss Weight ($\lambda_{\text{FM}}$) | FID (NFE=1, ↓) |
|---|---|
| 0.0–0.1 | Collapse |
| **0.1–10.0** | **3.17–3.22** |
| 10.0–64.0 | 3.32–4.97 |

| FM Loss Weight ($\lambda_{\text{FM}}$) | FID (NFE=1, ↓) |
|---|---|
| 0.0-1e-8 | Collapse |
| 1e-8–1e-4 | 2.90-5.88 |
| **1e-4–10.0** | **2.90–3.02** |
| 10.0–64.0 | 3.02–4.58 |

### 4.3.1 TRAINING DYNAMICS OF FACM

We analyze the training dynamics by plotting the total gradient norm under different configurations in Figure 3. Figure 3(a) clearly shows that removing the FM objective leads to catastrophic gradient spikes, after which the model's output immediately degenerates into pure noise (mode collapse). This confirms our hypothesis that a pure consistency gradient can trap the model in a local optimum where it sacrifices endpoint fidelity in pursuit of global consistency. Figure 3(b) further illustrates the effect of our auxiliary techniques. While each individually helps to suppress the gradient norm compared to removing them all, their combined use in our baseline model achieves the lowest and most stable gradient profile, demonstrating their synergistic effect in stabilizing the training process.

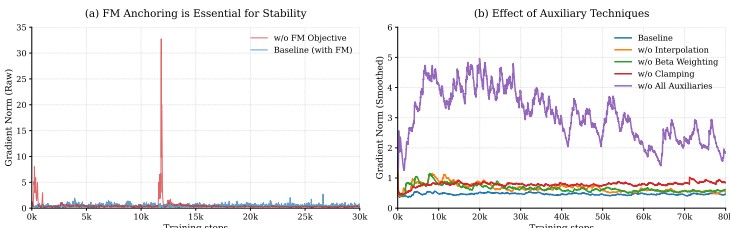

Figure 3: (a) The raw gradient norm for a pure CM (w/o FM Objective) shows an instantaneous spike leading to collapse, while our baseline remains stable. (b) The smoothed gradient norm for ablations of auxiliary techniques. Removing any single technique increases instability.

### 4.3.2 SCALABILITY ON A 14B TEXT-TO-IMAGE MODEL

Scaling continuous-time consistency models to billion-parameter scales presents a significant challenge due to the Jacobian-vector product (JVP) computation. While recent Differential Derivation Equation (DDE) methods (Sun et al., 2025; Wang et al., 2025) can yield results comparable to JVP on models up to 1B parameters, we observe that they exhibit significant deviation on larger models like our 14B setup. In such cases, their computed derivatives become nearly orthogonal or even opposed to the JVP result, indicating an inherent error accumulation that hinders further scalability. To address this, our memory-efficient **Chain-JVP** provides an accurate and scalable solution. To demonstrate its effectiveness, we applied FACM to distill the 14B parameter Wan 2.2 model. This process successfully accelerated inference from $2 \times 40$ steps to just 2-8 steps. For the experiment, we used a pre-trained Wan 2.2 Text-to-Video (T2V) model (Wan et al., 2025) as a teacher on an in-house Text-to-Image (T2I) dataset (Despite being a T2V model, Wan 2.2 has strong image generation capability from mixed image-video pre-training.). Furthermore, we adapt the model's self-attention and cross-attention mechanisms to be compatible with JVP computation, following the formulation of (Lu & Song, 2024). This adaptation also addresses the correctness of differentiation

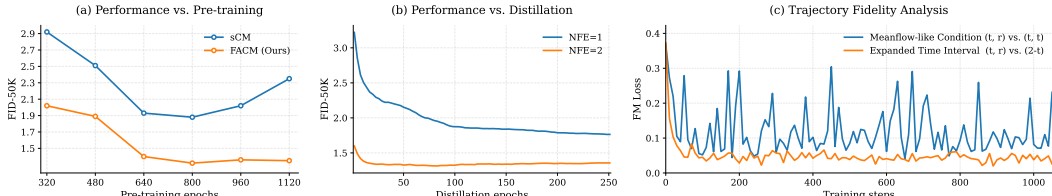

Figure 4: (a) Performance of student models (NFE=2) vs. teacher FM model pre-training epochs. (b) Performance vs. distillation epochs. (c) Trajectory fidelity analysis of 14B Wan 2.2 model via flow matching loss. Apart from the conditioning method, all other settings were the same.

for variable-length sequences and with bf16 precision. Our visualizations for this experiment are provided in Figure 1 and Appendix A.10, including comparison against the baseline model, as well as the FLUX.1-Dev (Labs, 2024a) and the FLUX.1-Schnell models (Labs, 2024b).

### 4.4 FROM CONSISTENCY MODELS TO FLOW MAPPING MODELS

Recent work has increasingly emphasized the advantages of Flow Mapping, where the model learns to predict the average velocity from an arbitrary time $t$ to another time $r$ (Sabour et al., 2025; Wang et al., 2025; Geng et al., 2025; Boffi et al., 2025). Flow Mapping requires the model to ensure that the derivative of $f_\theta(x_t, t, r) = x_t + (r - t)F_\theta(x_t, t, r)$ is zero. This formulation is an extension of consistency models along the trajectory, demanding that the model's prediction, $f_\theta(x_t, t, r)$, remains consistent over any time interval $[0, r]$ (detailed in Appendix A.6). We found through experiments on Wan 2.2 that FACM can be easily adapted to be compatible with the Flow Mapping formulation. This is achieved simply by changing $c_{\text{CM}}$ from $(t)$ to $(t, r)$ through zero-initialized time embedder and projection modules, while $c_{\text{FM}}$ is maintained in the separate, expanded time domain. As illustrated in Figure 4(c), the Expanded Time Interval strategy allows the Flow Mapping to be more stably anchored to the teacher's trajectory. If the prediction of the instantaneous velocity field is treated merely as a marginal case of the Auxiliary Time Condition (e.g., $r = t$), the FM loss becomes highly unstable, even when increasing the sampling proportion of $t = r$ as is done in MeanFlow. The consistently lower FM loss for the FACM condition demonstrates the superior trajectory fidelity achieved by our decoupled training strategy.

## 5 LIMITATIONS AND FUTURE WORK

Our work highlights two primary areas for future research. First, on large-scale models, a performance gap persists between samples generated in minimal steps (e.g., 1-2) and those requiring more steps (e.g., 8). Bridging this gap by enhancing the model's expressiveness in the ultra-few-step regime is a key challenge. Second, while our Chain-JVP method successfully mitigates the memory bottleneck of the Jacobian-vector product, its computational overhead remains a concern. Optimizing its efficiency is crucial for improving training throughput. Additionally, we found that our acceleration model, even when fine-tuned exclusively on T2I data, can directly accelerate T2V synthesis by targeting only the low-noise diffusion steps (SNR $\leq \frac{\text{SNR}_{\text{min}}}{2}$), all without introducing flickering or detail loss. This could inform future work on efficient, high-fidelity video synthesis.

## 6 CONCLUSION

In this work, we introduce the FACM, a strategy that addresses the instability of continuous-time CMs by anchoring the network to the underlying instantaneous velocity field with a Flow Matching loss. The core of this **Flow-Anchoring** approach is an expanded time interval strategy that unifies optimization for a single model via a continuous target, while functionally decoupling the anchoring and shortcut tasks to ensure high-fidelity and stability. Our method achieves new state-of-the-art FIDs on both ImageNet 256×256 (1.70 at NFE=1 and 1.32 at NFE=2) and CIFAR-10 (2.69 at NFE=1 and 1.87 at NFE=2). Furthermore, our **Chain-JVP** overcomes FSDP scalability bottlenecks, enabling us to accelerate a 14B model's inference from 2×40 to 2-8 steps.

ACKNOWLEDGMENTS

This work was supported in part by the National Natural Science Foundation of China under Grant 62472399.

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

## A APPENDIX

### A.1 THEORETICAL ANALYSIS: STABILITY AND CONVERGENCE OF FACM

This section provides a detailed mathematical derivation of the stability and convergence properties of FACM, complementing the intuitive discussion in Section 3.

We use the same notation as in the main text:

- Student network (online model): $\boldsymbol{F}_\theta(\boldsymbol{x}_t, t)$
- Stop-gradient copy: $\boldsymbol{F}_{\theta^-}(\boldsymbol{x}_t, t) = \mathrm{sg}(\boldsymbol{F}_\theta(\boldsymbol{x}_t, t))$
- Instantaneous velocity field (FM anchor): $\boldsymbol{v}(\boldsymbol{x}_t, t)$
- CM operator (Consistency target): $\mathcal{T}[\boldsymbol{F}] = \boldsymbol{v} + (1-t)\frac{\mathrm{d}\boldsymbol{F}}{\mathrm{d}t}$

#### A.1.1 STABILITY ANALYSIS: PREVENTION OF GRADIENT EXPLOSION

**Step 1.1: Structural form of the CM gradient.** The CM loss is

$$\mathcal{L}_{\mathrm{CM}}(\theta) = \mathbb{E}_{\boldsymbol{x}_t, t}\left[\frac{1}{2}\big\|\boldsymbol{F}_\theta(\boldsymbol{x}_t, t) - \boldsymbol{v}_{\mathrm{tar}}(\boldsymbol{x}_t, t)\big\|^2\right], \tag{16}$$

where $\boldsymbol{v}_{\mathrm{tar}}$ is the CM target derived from the operator $\mathcal{T}$ (Eq. 13 and Eq. 2 in the main text).

Differentiating w.r.t. $\theta$ yields the gradient in compact form:

$$
\begin{aligned}
\nabla_\theta \mathcal{L}_{\mathrm{CM}} &= \nabla_\theta \mathbb{E}_{\boldsymbol{x}_t, t}\left[\frac{1}{2}\big\|\boldsymbol{F}_\theta(\boldsymbol{x}_t, t) - \boldsymbol{v}_{\mathrm{tar}}(\boldsymbol{x}_t, t)\big\|^2\right] \\
&= \mathbb{E}_{\boldsymbol{x}_t, t}\left[\nabla_\theta\left(\frac{1}{2}(\boldsymbol{F}_\theta - \boldsymbol{v}_{\mathrm{tar}})^\top(\boldsymbol{F}_\theta - \boldsymbol{v}_{\mathrm{tar}})\right)\right] \\
&= \mathbb{E}_{\boldsymbol{x}_t, t}\left[(\nabla_\theta \boldsymbol{F}_\theta - \nabla_\theta \boldsymbol{v}_{\mathrm{tar}})^\top(\boldsymbol{F}_\theta - \boldsymbol{v}_{\mathrm{tar}})\right] \\
&\qquad \text{\footnotesize (Since $\boldsymbol{v}_{\mathrm{tar}}$ is a stop-gradient target, $\nabla_\theta \boldsymbol{v}_{\mathrm{tar}} = 0$)} \\
&= \mathbb{E}_{\boldsymbol{x}_t, t}\Big[\underbrace{\nabla_\theta \boldsymbol{F}_\theta(\boldsymbol{x}_t, t)^\top}_{\text{parameter sensitivity}} \cdot \underbrace{(\boldsymbol{F}_\theta(\boldsymbol{x}_t, t) - \boldsymbol{v}_{\mathrm{tar}}(\boldsymbol{x}_t, t))}_{\text{prediction error } \boldsymbol{e}}\Big],
\end{aligned} \tag{17}
$$

where $\nabla_\theta \boldsymbol{F}_\theta$ denotes the Jacobian of $\boldsymbol{F}_\theta$ w.r.t. the parameters $\theta$, and we write

$$\boldsymbol{e}(\boldsymbol{x}_t, t; \theta) := \boldsymbol{F}_\theta(\boldsymbol{x}_t, t) - \boldsymbol{v}_{\mathrm{tar}}(\boldsymbol{x}_t, t). \tag{18}$$

Taking norms and using $\|A^\top b\| \le \|A\|_{\mathrm{op}} \|b\|$, we obtain

$$\big\|\nabla_\theta \mathcal{L}_{\mathrm{CM}}\big\| \le \mathbb{E}_{\boldsymbol{x}_t, t}\Big[\big\|\nabla_\theta \boldsymbol{F}_\theta(\boldsymbol{x}_t, t)\big\|_{\mathrm{op}} \cdot \|\boldsymbol{e}(\boldsymbol{x}_t, t; \theta)\|\Big]. \tag{19}$$

Thus, the CM gradient norm is governed by two independent factors:

- the *prediction error* $\boldsymbol{e}$, which determines the basic scale and direction of the gradient;
- the *parameter sensitivity* $\nabla_\theta \boldsymbol{F}_\theta$, which acts as a multiplicative amplifier.

**Step 1.2: Decomposition of the error term.** Recall that the CM operator for a general field $\boldsymbol{F}$ is

$$\boldsymbol{v}_{\mathrm{tar}}(\boldsymbol{x}_t, t) = \boldsymbol{v}(\boldsymbol{x}_t, t) + (1-t)\Big(\partial_t \boldsymbol{F}_{\theta^-}(\boldsymbol{x}_t, t) + \nabla_{\boldsymbol{x}_t}\boldsymbol{F}_{\theta^-}(\boldsymbol{x}_t, t) \cdot \boldsymbol{v}(\boldsymbol{x}_t, t)\Big). \tag{20}$$

The error of the online model $\boldsymbol{F}_\theta$ relative to this target then decomposes as

$$\boldsymbol{e} = \underbrace{(\boldsymbol{F}_\theta - \boldsymbol{v})}_{\text{function deviation}} - \underbrace{(1-t)\frac{\partial \boldsymbol{F}_{\theta^-}}{\partial t}}_{\text{time derivative term}} - \underbrace{(1-t)\nabla_{\boldsymbol{x}_t}\boldsymbol{F}_{\theta^-} \cdot \boldsymbol{v}}_{\text{JVP (spatial Jacobian)}}. \tag{21}$$

Hence, the size of $\boldsymbol{e}$ is governed by the first-order spatio-temporal derivatives of the (stop-gradient) network $\boldsymbol{F}_{\theta^-}$.

**Step 1.3: FACM's stabilization mechanism.** FACM combines Flow Matching (FM) with the CM objective, using shared parameters $\theta$, and thereby stabilizes both factors in Eq. (19).

**(1) Lipschitz supervision via Flow Matching.** The FM loss (Eq. 9) trains $\boldsymbol{F}_\theta(\boldsymbol{x}_t, c_{\mathrm{FM}})$ to match the instantaneous velocity field $\boldsymbol{v}(\boldsymbol{x}_t, t)$, which is a bounded ground-truth function that does not depend on $\theta$ and is Lipschitz in $(\boldsymbol{x}_t, t)$. For standard architectures, the Lipschitz constant of $\boldsymbol{F}_\theta$ with respect to its inputs is determined only by the spectral norms of the weight matrices and the activation Lipschitz constants, all of which are shared across time conditions $t$ and $2 - t$. Minimizing the FM loss therefore keeps these spectral norms in a moderate range and induces a *global* Lipschitz bound

$$\left\| \nabla_\theta \boldsymbol{F}_\theta(\boldsymbol{x}_t, c) \right\|_{\mathrm{op}} \leq L_{\mathrm{net}} \tag{22}$$

for all $(\boldsymbol{x}_t, c)$ in the training domain, including both the FM branch ($c = 2 - t$) and the CM branch ($c = t$). Here, $L_{\mathrm{net}}$ represents the Lipschitz constant of the network, and $\| \cdot \|_{\mathrm{op}}$ denotes the spectral norm (operator norm). In contrast, pure CM supervises $\boldsymbol{F}_\theta$ with a self-referential target. To satisfy the consistency boundary condition (i.e., mapping $\boldsymbol{x}_t$ to $\boldsymbol{x}_1$ as $t \to 1$) under the parameterization $\boldsymbol{x}_t + (1 - t)\boldsymbol{F}_\theta(\boldsymbol{x}_t, t)$, the network output $\boldsymbol{F}_\theta$ is implicitly forced to approximate the average velocity $(\boldsymbol{x}_1 - \boldsymbol{x}_t)/(1 - t)$, which blows up as $t \to 1$. The CM target is therefore a dynamic, potentially unbounded prediction, whereas FM always provides a bounded ground-truth target.

**(2) Bounding the error via FM-anchored supervision.** Next, we show that the prediction error term $\boldsymbol{e}$ in the gradient (Eq. 19) remains bounded under FACM. For clarity, we focus on the deviation between the online model and the FM anchor, $\boldsymbol{F}_\theta(\boldsymbol{x}_t, t)$ vs. $\boldsymbol{v}(\boldsymbol{x}_t, t)$, and make explicit use of the *expanded time interval* strategy (Figure 2), where FM uses the condition $c_{\mathrm{FM}} = 2 - t$ but predicts the same physical velocity $\boldsymbol{v}(\boldsymbol{x}_t, t)$.

We begin with the triangle inequality:

$$\left\| \boldsymbol{F}_\theta(\boldsymbol{x}_t, t) - \boldsymbol{v}(\boldsymbol{x}_t, t) \right\| \leq \underbrace{\left\| \boldsymbol{F}_\theta(\boldsymbol{x}_t, t) - \boldsymbol{F}_\theta(\boldsymbol{x}_t, 2 - t) \right\|}_{\text{temporal smoothness of } \boldsymbol{F}_\theta} + \underbrace{\left\| \boldsymbol{F}_\theta(\boldsymbol{x}_t, 2 - t) - \boldsymbol{v}(\boldsymbol{x}_t, t) \right\|}_{\text{FM error}}. \tag{23}$$

The first term measures how much the network output changes when the (time-related) condition goes from $t$ to $2 - t$ for the same spatial point $\boldsymbol{x}_t$. Using the fundamental theorem of calculus for the time coordinate, we write

$$\boldsymbol{F}_\theta(\boldsymbol{x}_t, t) - \boldsymbol{F}_\theta(\boldsymbol{x}_t, 2 - t) = \int_{2-t}^{t} \frac{\partial}{\partial \tau} \boldsymbol{F}_\theta(\boldsymbol{x}_t, \tau, \tilde{c}(\tau)) \, d\tau, \tag{24}$$

where $\tilde{c}(\tau)$ interpolates between the CM and FM conditions as $\tau$ varies. Taking norms and applying the triangle inequality gives

$$\left\| \boldsymbol{F}_\theta(\boldsymbol{x}_t, t) - \boldsymbol{F}_\theta(\boldsymbol{x}_t, 2 - t) \right\| \leq \left| \int_{2-t}^{t} \left\| \frac{\partial}{\partial \tau} \boldsymbol{F}_\theta(\boldsymbol{x}_t, \tau, \tilde{c}(\tau)) \right\| d\tau \right|. \tag{25}$$

Since FM constrains the spectral norms of the weights, the partial time derivative $\left\| \partial_\tau \boldsymbol{F}_\theta(\cdot) \right\|$ is bounded by a constant (of order $L_{\mathrm{net}}$) over the training domain. The integration interval has length at most 2, so

$$\left\| \boldsymbol{F}_\theta(\boldsymbol{x}_t, t) - \boldsymbol{F}_\theta(\boldsymbol{x}_t, 2 - t) \right\| \leq 2 L_{\mathrm{net}}. \tag{26}$$

The second term in Eq. (23) is precisely the FM error

$$\varepsilon_{\mathrm{FM}}(\boldsymbol{x}_t, t) := \boldsymbol{F}_\theta(\boldsymbol{x}_t, 2 - t) - \boldsymbol{v}(\boldsymbol{x}_t, t), \tag{27}$$

whose squared norm is minimized by $\mathcal{L}_{\mathrm{FM}}$ and thus has bounded variance. Combining these bounds, we obtain a uniform control of the function deviation:

$$\left\| \boldsymbol{F}_\theta(\boldsymbol{x}_t, t) - \boldsymbol{v}(\boldsymbol{x}_t, t) \right\| \leq 2 L_{\mathrm{net}} + \left\| \varepsilon_{\mathrm{FM}}(\boldsymbol{x}_t, t) \right\|. \tag{28}$$

Finally, because the same spectral-norm constraints apply to all weight matrices, the spatial Jacobian $\nabla_{\boldsymbol{x}_t} \boldsymbol{F}_{\theta^-}$ and time derivative $\partial_t \boldsymbol{F}_{\theta^-}$ in Eq. (21) are also uniformly bounded by constants of order $L_{\mathrm{net}}$. Hence, all components of $\boldsymbol{e}$ remain bounded.

**Conclusion (Stability).**    Putting these results together, we see that under FACM:

- the parameter sensitivity is bounded: $\|\nabla_\theta \boldsymbol{F}_\theta\|_{\mathrm{op}} \le L_{\mathrm{net}}$;
- the prediction error $\boldsymbol{e}$ is uniformly bounded in norm by a constant depending on $L_{\mathrm{net}}$ and the FM error statistics.

Therefore, the CM gradient norm satisfies

$$\left\| \nabla_\theta \mathcal{L}_{\mathrm{CM}} \right\| \le C\, L_{\mathrm{net}}^2, \tag{29}$$

eliminating the gradient explosion observed in pure CM training.

### A.1.2   Convergence Proof

We provide a concise analysis showing that FACM ensures convergence by eliminating target singularities, bounding gradient variance, and enforcing alignment through shared parameters.

**Step 2.1: Mechanism of Variance Reduction.**    The pure consistency target implies predicting the average velocity

$$\overline{\boldsymbol{v}}(\boldsymbol{x}_t, t) = \frac{\boldsymbol{x}_1 - \boldsymbol{x}_t}{1 - t}. \tag{30}$$

As $t \to 1$, any variance $\sigma^2$ in the endpoint estimate $\boldsymbol{x}_1$ is amplified by $(1-t)^{-2}$, so

$$\mathrm{Var}[\overline{\boldsymbol{v}}(\boldsymbol{x}_t, t) \mid t] = \frac{\sigma^2}{(1-t)^2}, \tag{31}$$

which makes the pure-CM shortcut objective ill conditioned near the data endpoint. FACM mitigates this by using a relaxed target

$$\boldsymbol{v}_{\mathrm{tar}}(\boldsymbol{x}_t, t) = (1 - \alpha(t))\boldsymbol{F}_{\theta^-}(\boldsymbol{x}_t, t) + \alpha(t)\,\overline{\boldsymbol{v}}(\boldsymbol{x}_t, t). \tag{32}$$

In practice we use the schedule $\alpha(t) = 1 - t^{0.5}$ (Sec. 3). Writing $t = 1 - \varepsilon$ with $0 < \varepsilon \ll 1$ and using the Taylor expansion

$$t^{0.5} = (1 - \varepsilon)^{0.5} = 1 - \tfrac{1}{2}\varepsilon + \mathcal{O}(\varepsilon^2), \tag{33}$$

we obtain

$$\alpha(t) = 1 - t^{0.5} = \tfrac{1}{2}\varepsilon + \mathcal{O}(\varepsilon^2) = \tfrac{1}{2}(1 - t) + \mathcal{O}\big((1-t)^2\big). \tag{34}$$

Hence $\alpha(t)$ is asymptotically proportional to $(1 - t)$ and

$$\lim_{t \to 1} \frac{\alpha(t)}{1 - t} = \tfrac{1}{2}, \tag{35}$$

so the factor $\alpha(t)$ cancels the $(1 - t)^{-1}$ singularity in the average-velocity term up to a constant. Consequently $\mathrm{Var}[\boldsymbol{v}_{\mathrm{tar}}(\boldsymbol{x}_t, t) \mid t]$ remains uniformly bounded over $t \in [0, 1]$, providing the core mechanism for variance reduction in FACM.

**Step 2.2:  Bounded and Reduced Gradient Variance.**    Locally, $L_{\mathrm{norm}}$ behaves like a rescaled squared $\ell_2$ loss, so the CM gradient for one sample $(\boldsymbol{x}_t, t)$ satisfies

$$\left\| \nabla_\theta \ell_{\mathrm{CM}}(\theta; \boldsymbol{x}_t, t) \right\| \lesssim \beta(t)\, \|\nabla_\theta \boldsymbol{F}_\theta(\boldsymbol{x}_t, t)\|_{\mathrm{op}}\, \|\boldsymbol{F}_\theta(\boldsymbol{x}_t, t) - \boldsymbol{v}_{\mathrm{tar}}(\boldsymbol{x}_t, t)\|^2. \tag{36}$$

Using the stability bound $\|\nabla_\theta \boldsymbol{F}_\theta\|_{\mathrm{op}} \le L_{\mathrm{net}}$ and the uniform boundedness of $\boldsymbol{v}_{\mathrm{tar}}$, we obtain a finite second-moment (and hence variance) bound

$$\mathbb{E}\big[\|\widehat{\nabla}_\theta \mathcal{L}_{\mathrm{CM}}\|^2\big] \;\lesssim\; L_{\mathrm{net}}^2\, \mathbb{E}_t\Big[\beta(t)^2\, \mathbb{E}_{\boldsymbol{x}_t}\big[\|\boldsymbol{F}_\theta(\boldsymbol{x}_t, t) - \boldsymbol{v}_{\mathrm{tar}}(\boldsymbol{x}_t, t)\|^2 \mid t\big]\Big]. \tag{37}$$

Here the *boundedness* follows from the variance-reduction mechanism in the previous paragraph, which shows that $\mathrm{Var}[\boldsymbol{v}_{\mathrm{tar}}(\boldsymbol{x}_t, t) \mid t]$ is uniformly bounded in $t$, together with the global Lipschitz bound $\|\nabla_\theta \boldsymbol{F}_\theta\|_{\mathrm{op}} \le L_{\mathrm{net}}$ from Step 1.3: even if $\beta(t) \equiv 1$, the right-hand side is finite because both the Jacobian norm and the target variance are controlled. The role of $\beta(t) \in [0, 1]$ is to *further reduce* the integrated variance: for any non-negative function $q(t)$,

$$\mathbb{E}_t[\beta(t)^2 q(t)] \le \mathbb{E}_t[q(t)], \tag{38}$$

with strict inequality whenever $\beta(t) < 1$ on a set of non-zero measure. Since $q(t) = \mathbb{E}_{\boldsymbol{x}_t}\|\boldsymbol{F}_\theta(\boldsymbol{x}_t, t) - \boldsymbol{v}_{\mathrm{tar}}(\boldsymbol{x}_t, t)\|^2 \mid t$ is typically largest near the data endpoint $t \approx 1$, choosing a decaying schedule (e.g., cosine) for $\beta(t)$ suppresses precisely those high-variance contributions, yielding a strictly lower integrated gradient variance than the pure-CM case.

**Step 2.3: Alignment via Shared Parameters.** The expanded time interval $[0, 2]$ creates a natural synchronization mechanism between the FM and CM tasks. In the high-SNR region ($t \approx 1$), the schedules $\alpha(t)$ and $\beta(t)$ suppress the CM gradients, so parameter updates there are dominated by the FM branch, which supervises $\boldsymbol{F}_\theta(\boldsymbol{x}_t, 2-t)$ to match the instantaneous velocity field $\boldsymbol{v}(\boldsymbol{x}_t, t)$. Because both branches share the same parameters and the stability analysis above bounds the discrepancy between $\boldsymbol{F}_\theta(\boldsymbol{x}_t, t)$ and $\boldsymbol{F}_\theta(\boldsymbol{x}_t, 2-t)$, this FM supervision implicitly guides $\boldsymbol{F}_\theta(\boldsymbol{x}_t, t)$ to align with the underlying velocity field. Combined with the reduced gradient variance from Step 2.2, this shared-parameter coupling explains the empirically faster and more stable convergence of FACM compared to pure CM training.

## A.2 ON TOTAL DERIVATIVES

In this paper, for a network $N(\boldsymbol{x}_t, \mathcal{C}(t))$ (e.g., $\boldsymbol{F}_\theta$), its total derivative along the trajectory $\boldsymbol{x}_t(t) = (1-t)\boldsymbol{x}_0 + t\boldsymbol{x}_1$ (with $\boldsymbol{v} = \frac{\mathrm{d}\boldsymbol{x}_t}{\mathrm{d}t} = \boldsymbol{x}_1 - \boldsymbol{x}_0$) with respect to $t$ is given by the chain rule:

$$\frac{\mathrm{d}N(\boldsymbol{x}_t, \mathcal{C}(t))}{\mathrm{d}t} = \frac{\partial N}{\partial \boldsymbol{x}_t}\boldsymbol{v} + \nabla_{\mathcal{C}} N \cdot \frac{\mathrm{d}\mathcal{C}(t)}{\mathrm{d}t}. \tag{39}$$

The term $\frac{\mathrm{d}\boldsymbol{F}_{\theta^-}(\boldsymbol{x}_t, c_{\mathrm{CM}})}{\mathrm{d}t}$ is computed for the CM task. Depending on the implementation strategy (Sec. 3.3.2), the conditioning $c_{\mathrm{CM}}$ can be $t$ or a tuple $(t, 1)$. In both cases, its derivative with respect to $t$ is effectively 1 for the time-dependent component and 0 for any constant component. Therefore, the calculation simplifies to:

$$\frac{\mathrm{d}N(\boldsymbol{x}_t, c_{\mathrm{CM}})}{\mathrm{d}t} \approx \frac{\partial N}{\partial \boldsymbol{x}_t}\boldsymbol{v} + \frac{\partial N}{\partial t}, \tag{40}$$

where $\frac{\partial N}{\partial t}$ denotes the partial derivative with respect to the explicit time argument(s) encoded in the conditioning.

## A.3 NORM L2 LOSS

The CM loss component uses a norm L2 loss to improve stability against outliers. For a model prediction $\boldsymbol{p}$ and a target $\boldsymbol{y}$, let the per-sample squared error be $e = \|\boldsymbol{p} - \boldsymbol{y}\|_2^2$. The loss is then calculated as:

$$L_{\mathrm{norm}}(\boldsymbol{p}, \boldsymbol{y}) = \frac{e}{\sqrt{e + c^2}} \tag{41}$$

where $c$ is a small constant. This formulation is equivalent to the adaptive L2 loss proposed in MeanFlow (Geng et al., 2025) with $p = 0.5$, and behaves similarly to a Huber loss, being robust to large errors.

## A.4 EXPERIMENTAL DETAILS

**(a) Pre-training Strategy.** Our teacher models are standard Flow Matching models. While FACM distillation works perfectly with a standard, single-condition pre-trained teacher, we find that convergence can be accelerated by first familiarizing the teacher with our dual-task conditioning. This optional adaptation can be achieved either by pre-training from scratch with a mixed-conditioning objective (i.e., replacing the standard time conditioning with our FM-specific formats for 50% of samples) or by briefly fine-tuning a pre-trained FM model with this objective for a few epochs. Furthermore, to prevent sporadic *NaN* losses during pre-training, all our LightningDiT implementations incorporate Query-Key Normalization (QKNorm), following updates in the official repository.

**(b) Sampling Strategy.** Our multi-step sampling (NFE $\geq 2$) follows a standard iterative refinement process. For an $N$-step generation, we use a simple schedule of $N$ equally spaced timesteps $t_i = (i-1)/N$ for $i = 1, \ldots, N$. The process starts with pure noise $\boldsymbol{x}_0$. At each step $i$, we first compute a one-step prediction $\hat{\boldsymbol{x}}_1$ using the model's output $\boldsymbol{F}_\theta$: $\hat{\boldsymbol{x}}_1 = \boldsymbol{x}_{t_i} + (1 - t_i)\boldsymbol{F}_\theta(\boldsymbol{x}_{t_i}, c_{\mathrm{CM}})$. If it is not the final step, we generate the input for the next step, $\boldsymbol{x}_{t_{i+1}}$, by linearly interpolating between the predicted endpoint and a new noise sample, consistent with the OT-FM framework:

$$\boldsymbol{x}_{t_{i+1}} = t_{i+1}\hat{\boldsymbol{x}}_1 + (1 - t_{i+1})z_i, \quad \text{where } z_i \sim \mathcal{N}(0, I). \tag{42}$$

The final output is the prediction from the last timestep, $t_N$.

**(c) Reproduction Details.** At the time of our main ablations, the official codebases for Mean-Flow (Geng et al., 2025) and sCM (Lu & Song, 2024) were not yet available. A JAX implementation of MeanFlow was later released, but without a reproducible configuration for its SOTA results. For a controlled and fair comparison, we therefore implemented PyTorch reproductions under the exact same environment, teacher, and hyperparameters across methods. Our MeanFlow reproduction follows its two-time-variable conditioning and log-normal time sampling; following the from-scratch regime, we set $t = r$ with a 75% probability for optimal performance. In the distillation setting, this configuration struggled to converge and was therefore not used. For sCM, we incorporated all necessary techniques described in their work, including pixel normalization, tangent warmup, tangent normalization, and adaptive weighting, to ensure stable training. We did not use the TrigFlow proposed in sCM, as we believe the specific flow construction is orthogonal to building continuous-time consistency models. We will release our reproductions alongside our code to ensure full reproducibility.

**(d) Classifier-Free Guidance in Distillation.** When distilling a teacher model that supports classifier-free guidance (CFG), we compute both the conditional velocity $\boldsymbol{v}_{\text{cond}}$ and unconditional velocity $\boldsymbol{v}_{\text{uncond}}$ from the teacher, and construct the target as

$$\boldsymbol{v} = \boldsymbol{v}_{\text{uncond}} + w \cdot (\boldsymbol{v}_{\text{cond}} - \boldsymbol{v}_{\text{uncond}}), \tag{43}$$

where $w$ is the guidance weight. During training, the unconditional forward is computed with `torch.no_grad()`, adding less than 5% overhead, which is consistent with sCM and MeanFlow. To stabilize the high-noise region, we set a time threshold $t_{\text{low}}$ and disable guidance for $t < t_{\text{low}}$ (we use $t_{\text{low}}$=0.05 in our experiments). During pre-training, the null-token probability is 10%, and the condition is not dropped during distillation. At inference, FACM uses a single timestamp; even under CFG, each step requires only **one** NFE.

**(e) Time Sampling Schedule.** Following sCM (Lu & Song, 2024), the time $t \in [0, 1]$ is sampled according to a schedule that concentrates samples near the data endpoint ($t = 1$). We first sample a value $\sigma$ from a log-normal distribution, i.e., $\ln(\sigma) \sim \mathcal{N}(P_{\text{mean}}, P_{\text{std}}^2)$, and then compute $t$ as:

$$t = 1 - \frac{2}{\pi} \arctan(\sigma). \tag{44}$$

**(f) Weighting Functions.** For the CM loss component (Eq. 13), we find that the weighting functions $\alpha(t) = 1 - t^{0.5}$ and $\beta(t) = \cos(t \cdot \pi/2)$ provide an effective general solution. These functions are crucial for navigating the trade-off between ensuring endpoint quality (in high-SNR regions) and satisfying global consistency (in low-SNR regions).

### A.5 Discussion: From-Scratch Training vs. Distillation

While our method can be trained from scratch and achieves a competitive result (See Table 3), we identify the two-stage distillation paradigm as the more principled and practically superior approach. Attempting to learn both the anchor and the shortcut simultaneously from scratch introduces a "chicken-and-egg" problem, as the model must learn a shortcut based on a trajectory it has not yet accurately modeled. This creates an unstable "moving target" for optimization and incurs higher computational costs. In contrast, distillation from a pre-trained FM teacher provides a fixed, high-quality velocity field, offering a much more stable and well-defined learning objective. MeanFlow (Geng et al., 2025) also encounters this problem in the from-scratch setting, achieving its optimal performance only by having its objective degenerate to a Flow Matching task for a large portion of samples (e.g., 75%), which further validates our core thesis that a robust foundation in the velocity field is a prerequisite for learning stable shortcuts.

## A.6 FLOW MAPPING EQUIVALENCE DERIVATION

Let the Flow Mapping function be $f_\theta(\boldsymbol{x}_t, t, r) = \boldsymbol{x}_t + (r - t)\boldsymbol{F}_\theta(\boldsymbol{x}_t, t, r)$. The consistency condition $\frac{d}{dt}f_\theta(\boldsymbol{x}_t, t, r) = 0$ is equivalent to the learning objective for $\boldsymbol{F}_\theta$:

$$\frac{d}{dt}f_\theta(\boldsymbol{x}_t, t, r) = 0 \quad \Longleftrightarrow \quad \boldsymbol{v} - \boldsymbol{F}_\theta(\boldsymbol{x}_t, t, r) + (r - t)\frac{\mathrm{d}\boldsymbol{F}_\theta}{\mathrm{d}t} = 0$$

$$\Longleftrightarrow \quad \boldsymbol{F}_\theta(\boldsymbol{x}_t, t, r) = \boldsymbol{v} + (r - t)\frac{\mathrm{d}\boldsymbol{F}_\theta}{\mathrm{d}t}$$

Enforcing this for $t \in [0, r]$ implies that $f_\theta$ is constant over the interval, thus mapping any point $\boldsymbol{x}_t$ to the endpoint $\mathbf{x}_r$:

$$f_\theta(\boldsymbol{x}_t, t, r) \overset{t \in [0,r]}{=} f_\theta(\mathbf{x}_r, r, r) = \mathbf{x}_r$$

## A.7 HYPERPARAMETERS

Table 6: Key hyperparameters for our experiments.

| Hyperparameter | Value | Hyperparameter | Value | Cifar-10 Value |
|---|---|---|---|---|
| Optimizer | AdamW | Batch Size | 1024 | 128 |
| Learning Rate | 1e-4 | Time Sampling ($P_{\mathrm{mean}}, P_{\mathrm{std}}$) | (-0.8, 1.6) | (-1.0, 1.4) |
| Weight Decay | 0 | CFG Scale ($w$) | 1.75 | 1.0 |
| EMA Length ($\sigma_{\mathrm{rel}}$) | 0.2 | Flow Schedule | OT-FM | Simple-EDM |
| Norm L2 Loss $c$ | 1e-3 | Dropout | 0 | 0.2 |
| CFG $t_{\mathrm{low}}$ | 0.05 | AdamW Betas ($\beta_1, \beta_2$) | (0.9, 0.999) | (0.9, 0.99) |

## A.8 ABLATION ON THE COSINE SIMILARITY TERM

The FM loss in Eq. 9 includes a cosine similarity term, which we found to be beneficial for aligning with pre-trained VAE/DiT teachers whose features are trained with representation supervision. Across our ImageNet 256×256 experiments (NFE=1), removing this term consistently degrades FID by 0.1–0.2. We therefore keep it as a default component of the FM loss.

## A.9 COMPUTATIONAL COST AND RESOURCES

**Generation Latency.** On a single A100 GPU, our 2-step FACM sampler takes approximately 70.2 ms per image (including VAE decoding), versus 7062.9 ms for a standard 250-step Euler sampler, translating to roughly ~100× speed-up in wall-clock time.

**JVP Memory and Throughput.** Our Chain-JVP introduces no bias to the derivative and is embedded within the FSDP backend, so its speed matches a standard FSDP forward with differentiation. It reduces peak memory from an OOM error to ~72GB for a 14B-parameter model on 80GB A100s. For a 5B model, Chain-JVP with FlashAttention2 reduces peak memory from ~76GB to ~38GB.

**14B Distillation Resources.** We distilled the 14B model using $64 \times$ A100 GPUs. The NFE=8 results reported in the paper were obtained after 5000 steps with a batch size of 512, taking 73 hours. The same setting is reproducible on fewer GPUs via gradient accumulation and FSDP CPU Offload.

## A.10 ADDITIONAL VISUALIZATION

| FACM (Ours) 14B NFE=8 | Wan2.2 A14B NFE=40×2 | FLUX.1-Schnell 12B NFE=8 | FLUX.1-Dev 12B NFE=50×2 |

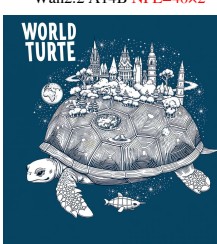 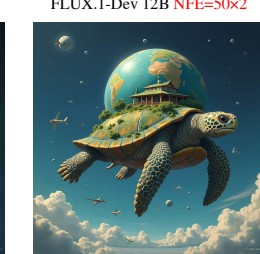

A detailed illustration of a "world turtle," a giant turtle carrying a whole fantasy world on its back, swimming through space.

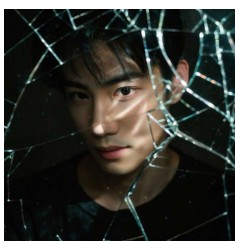 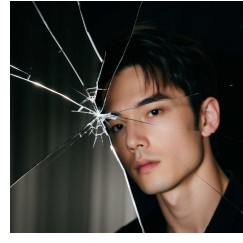 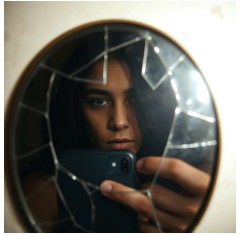 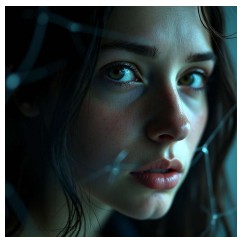

A close-up selfie in a cracked mirror, the flash highlighting the cracks and the subject's face, moody and introspective.

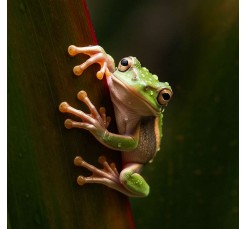 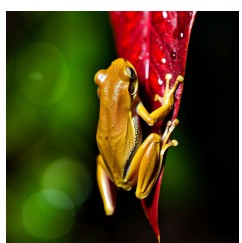 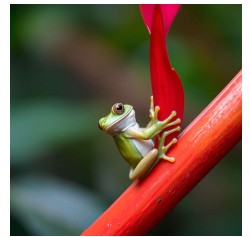 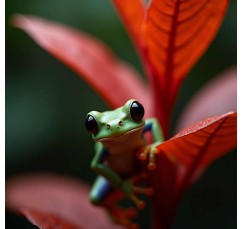

A tiny tree frog clinging to a vibrant red leaf, its skin glistening with moisture, rich jungle background bokeh.

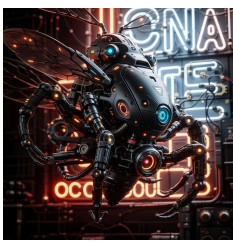 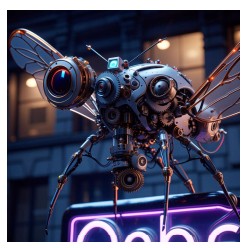 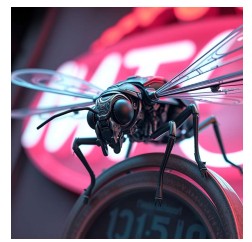 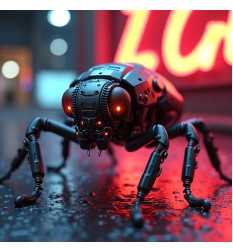

Intricate close-up of a mechanical insect drone, detailed gears and sensors, near a neon sign.

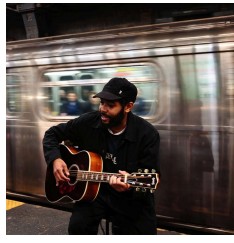 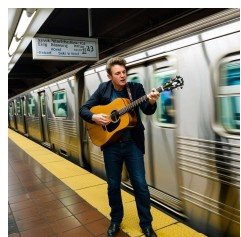 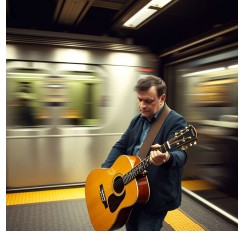 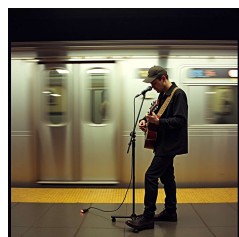

A musician playing a guitar in a New York City subway station, motion blur of the passing train in the background, authentic moment.

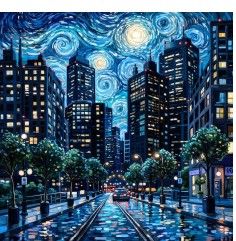 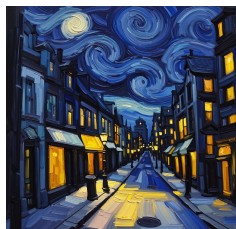 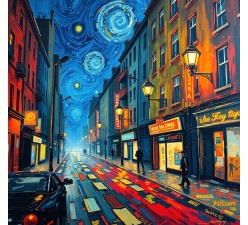 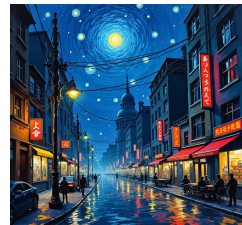

A cyberpunk city street at night, painted with thick, swirling impasto brushstrokes, in the style of Vincent van Gogh.

FACM (Ours) 14B NFE=8   Wan2.2 A14B NFE=40×2   FLUX.1-Schnell 12B NFE=8   FLUX.1-Dev 12B NFE=50×2

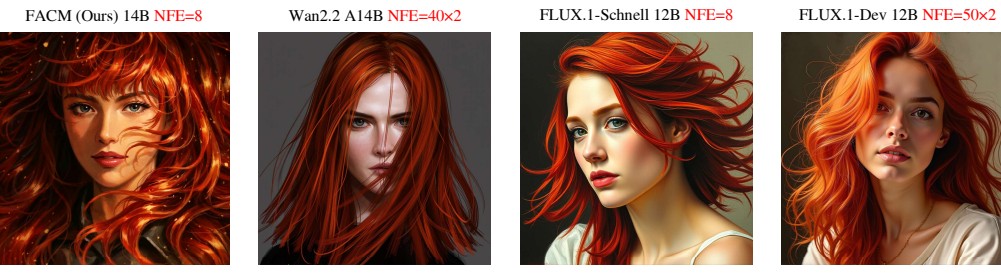

A portrait of a woman with fiery red hair, each strand a distinct, thick brushstroke, full of movement.

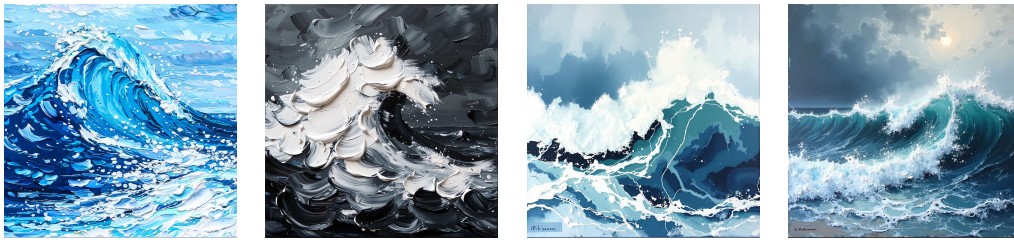

An expressive, thick impasto oil painting of a stormy seascape, waves crashing with heavy, textured white paint, palette knife.

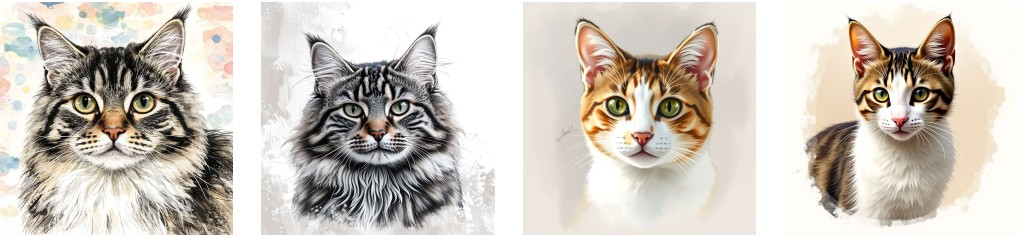

A portrait of a cat, its fur suggested with dry brush technique over a soft wash background.

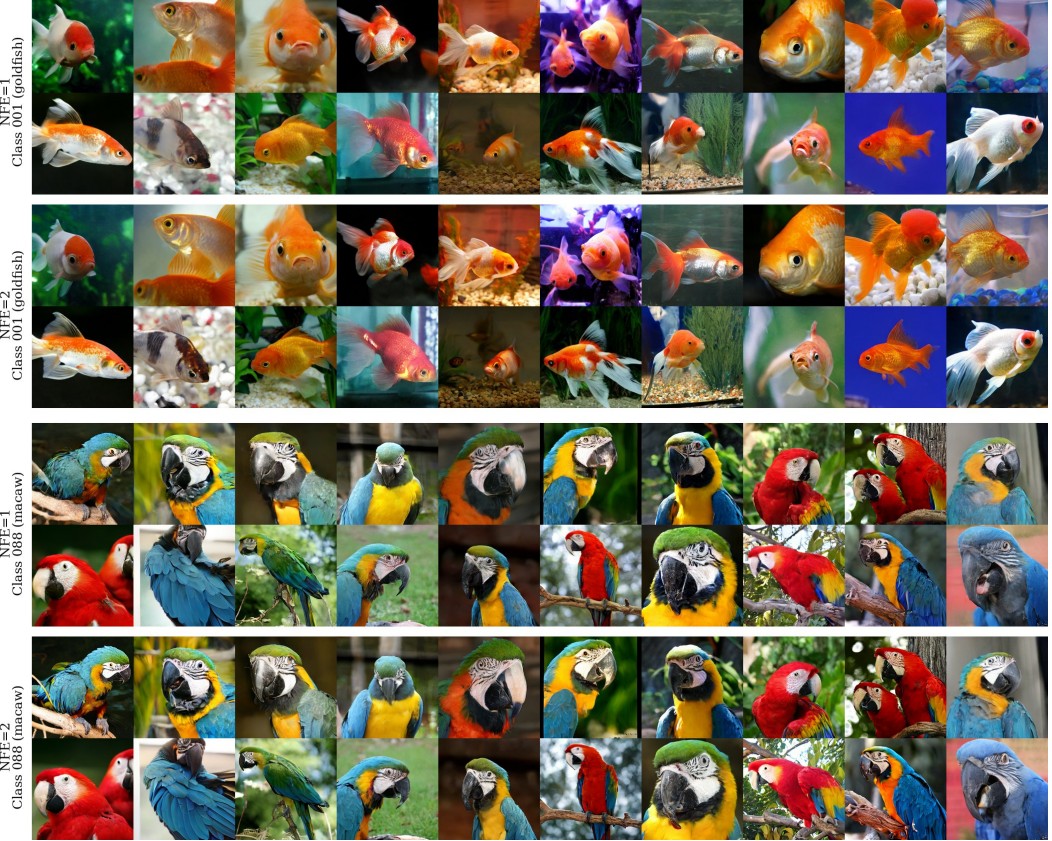

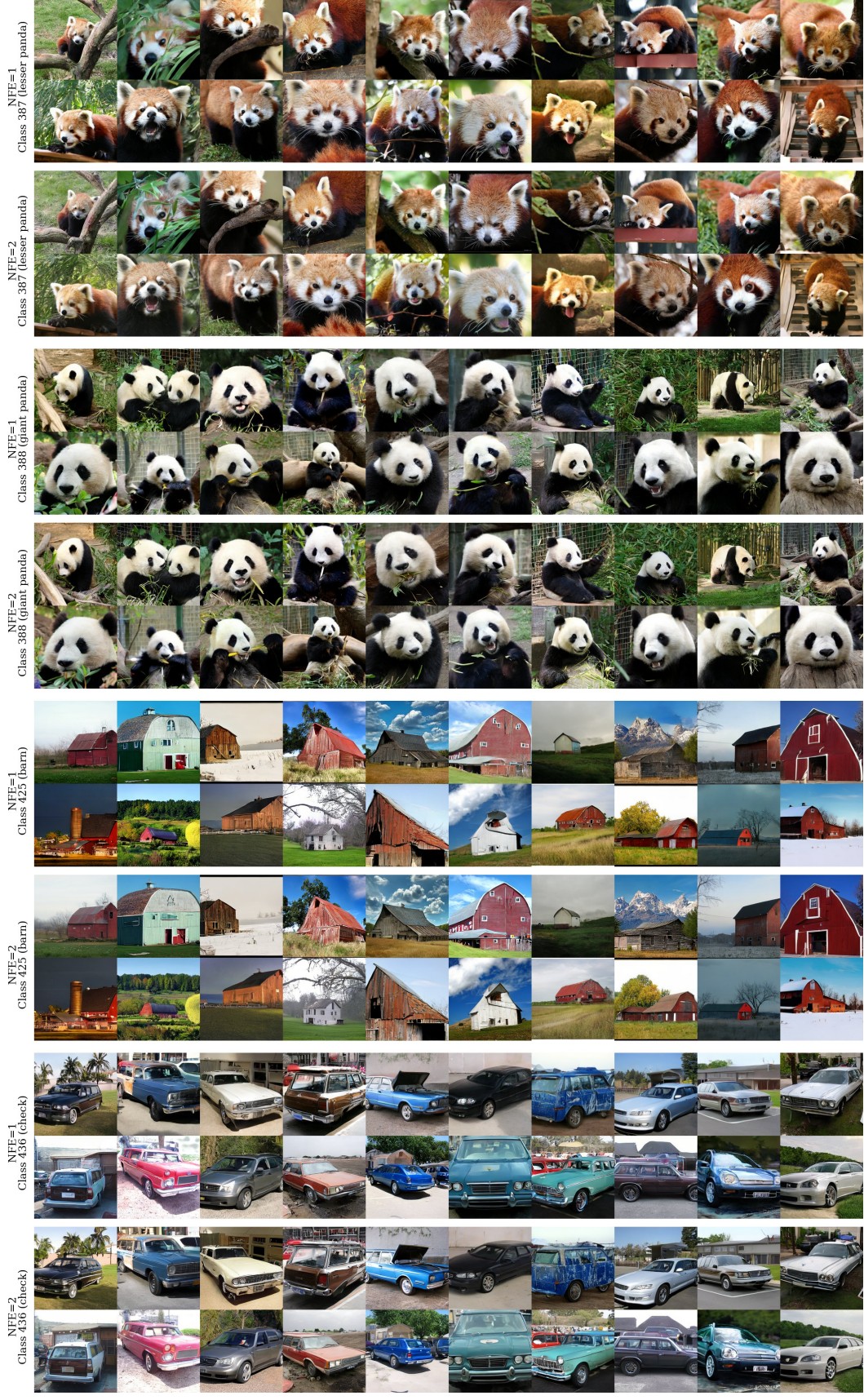

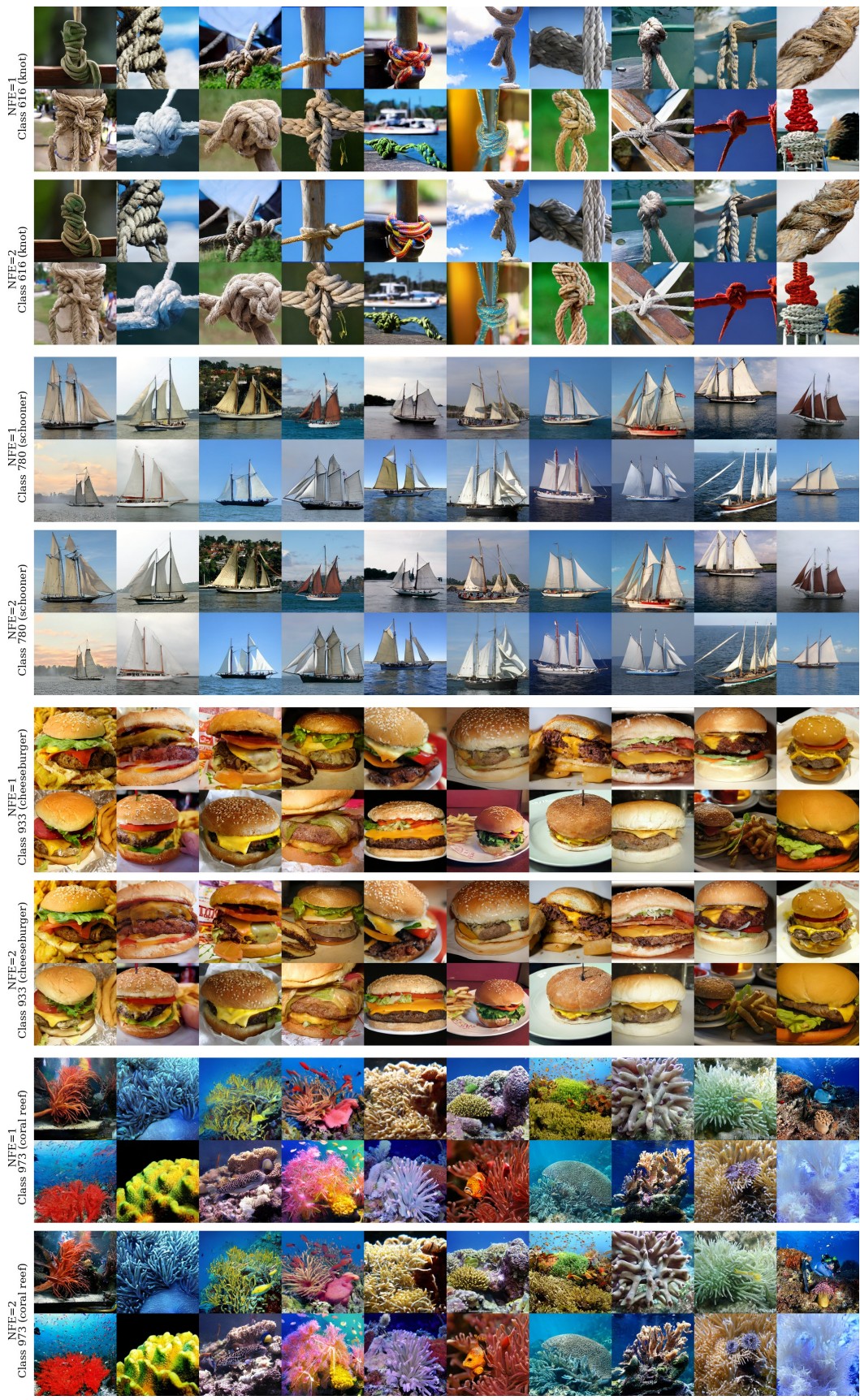

Uncurated T2I generation results of FACM 14B. The generations are based on a batch of randomly sampled prompts. The images from left to right are generated with different NFE: 2, 4, 6, and 8, respectively

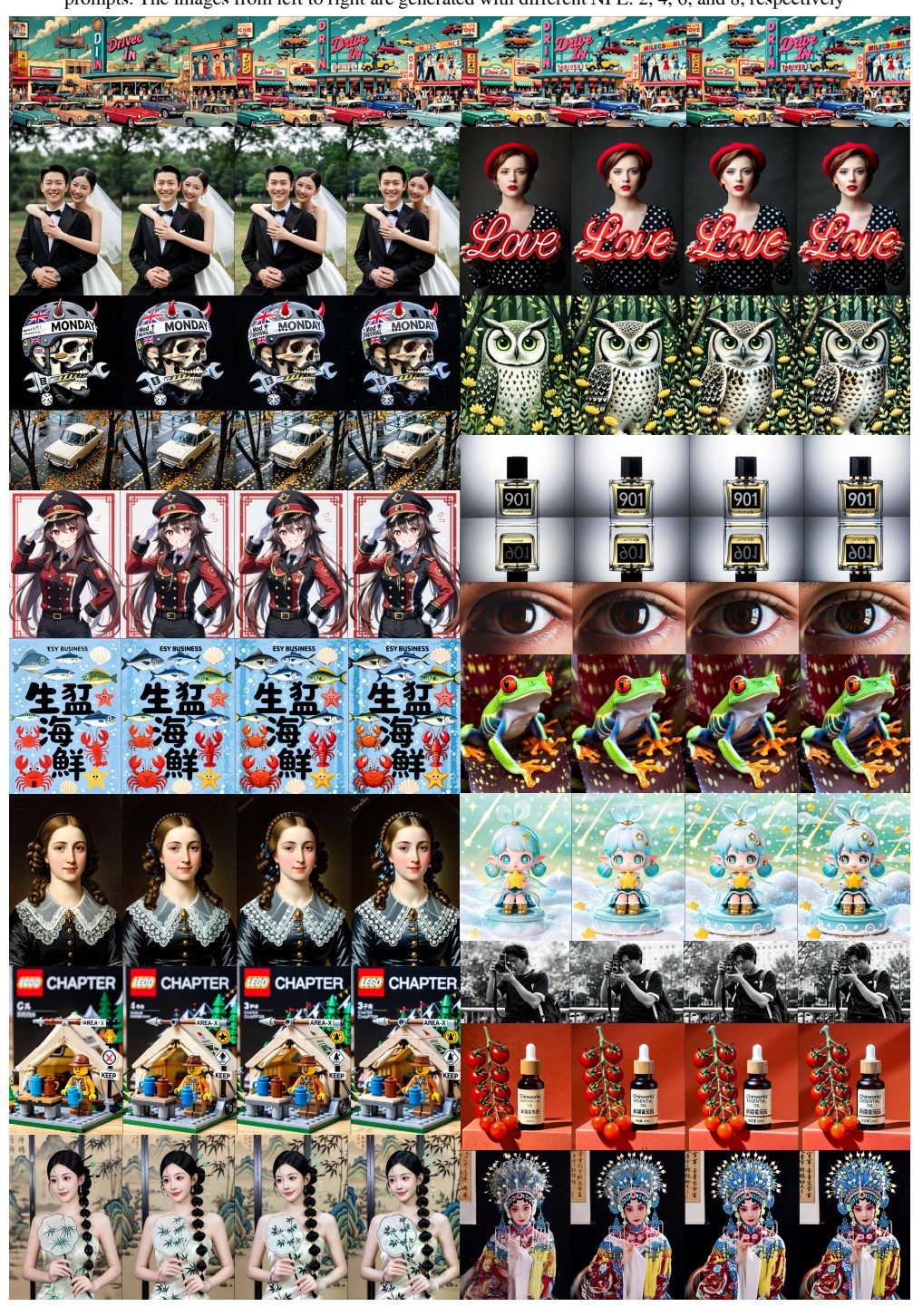

## A.11 PROMPTS FOR TEASER VISUALIZATIONS

The following are the text prompts used for the text-to-image synthesis examples shown in the top two rows of Figure 1.

- A soldier in tactical gear standing next to a modified desert-runner muscle car in a vast desert under a bright sun, digital art.
- A surrealist portrait where the person's face is a composite of various flowers and leaves.
- A portrait of a girl whose hair is made of flowing, colorful ink, watercolor style.
- Close-up of a tarot card, "The World", depicting a cyborg wreathed in stars.
- A painting of a time traveler's footprints through history, each print leading to a different era.
- A man with a worried expression looking out through the window, overcast lighting.
- A florist arranging a bouquet of fresh flowers, a beautiful combination of colors and scents.
- A woman in a corner of the library, surrounded by books, studying quietly.
- An artist in her studio, splattered with paint, staring intently at a large canvas, dramatic lighting, Rembrandt lighting.
- A street musician playing the cello in a European city square, a little girl stops to listen, touching moment.

## A.12 THE USE OF LARGE LANGUAGE MODELS (LLMS)

In accordance with ICLR 2026 policy, we report the use of a Large Language Model (LLM) during the preparation of this manuscript. We used a large language model as a writing assistant to help improve the grammar and clarity of our prose. Its role was strictly limited to proofreading; all scientific ideas, analyses, and conclusions presented are our own. The authors have reviewed the final text and take full responsibility for its content.

