# OpenReview forum: "FACM: Flow-Anchored Consistency Models"
_ICLR.cc/2026/Conference — ICLR 2026 Poster_

### Official Review · Reviewer_8cpy · 2025-10-27

**Soundness:** 3
**Presentation:** 3
**Contribution:** 3
**Rating:** 6
**Confidence:** 4

**Summary:**

The paper proposed FACM, a method for stabilizing and improving training of continuous consistency models. The method consists in combining the continuous consistency objective with a flow matching anchor, to encourage the model to learn the underlying flow, which results in improved performance and stability. The method further introduces a novel time embedding strategy and a JVP computation which allows scaling to models with billions of parameters. The method is validated on image generation benchmarks and text-to-image benchmarks and results in competitive performance, beating other one and few-step generative models.

**Strengths:**

The method is a simplification of continuous consistency models, by leveraging the features learned with flow-matching training. The novel objective is well motivated and effective, and achieves strong performance. The introduction of the scalable JVP computation is valuable for the research community.

**Weaknesses:**

The method uses now two forward passes, one of which requires JVP computation, so I am concerned about the training efficiency. Nothing about it is reported in the paper, but I think it is important to mention how slower the model gets when trained with FACM.
While simplified compared to sCM, the method still requires complex formulations and weighting functions, for which no ablation is provided.

As a minor note, the authors claim that the MeanFlow code is not available, while it was already by the time of the submission deadline.

**Questions:**

- Regarding CFG at sampling, does the model require only one NFE or two NFEs? And if it's the 1 NFE case, is the model only conditioned on the conditioning signal, or does it require extra parameters like in MeanFlow?
- Still regarding CFG training, how slower is the model, given that it requires one additional forward pass for $v_{cond}$? Also there does not seem to be any information regarding the probability of using the null token for the pretrained/online base model. In addition, $t_{low}$ is only mentioned as hyperparameter but never in the text.
- What's the reason behind the cosine similarity loss for the FM objective? I am not familiar with this term being used in the flow/score literature, is this part of the contribution or is it commonly used in other FM methods?
- Are the consistency and FM losses simply summed together? Do you use any weighting term or a percentage of the batch similar to Shortcut Models or MeanFlow?

---

> ### Author Response · Authors · 2025-11-21
> **Response to Reviewer 8cpy**
>
> **[W1] Training Efficiency**:
>
> Thank you for raising this important point. The CM portion of FACM has an identical memory footprint and training speed to sCM and MeanFlow. Per step, our method has one more backward pass (for the FM loss) than sCM, but two fewer non-gradient forward passes than MeanFlow, making our computational costs comparable. **However, the key advantage is in convergence speed. On ImageNet, our NFE=2 model surpasses the final FID of fully-trained sCM and MeanFlow in just 1k steps.** This efficiency scales to our 14B model, which acquires high-quality NFE=2/4/8 generation in just a few hundred fine-tuning steps (~8 hours), demonstrating a significant reduction in overall training cost.
>
> **[W2] Availability of MeanFlow Code**:
>
> Thank you for the correction. At the time we conducted our main ablations, MeanFlow was not yet open-sourced. When it was released, it was in JAX and did not include a reproducible configuration for its SOTA results. By using our own PyTorch reproduction, we could perform a more controlled comparison, using the exact same environment, hyperparameters, and teacher model to isolate the performance difference of the distillation algorithm itself, and our reproduced results were superior to those reported in their original paper. We believe this makes our comparison more rigorous.
>
> **[Q1, Q2] CFG at Sampling and Training**:
>
> During inference, **FACM uses a single timestamp.** A time in $[0, 1]$ activates the consistency sampler for fast generation (1-8 NFE), while a time in $[1, 2]$ uses the original pre-trained model's sampler for high-fidelity generation. Since the model has learned the CFG capability, each step only requires one NFE.
>
> During training, the extra forward pass for the unconditional output is performed with *torch.no_grad()*, **slowing down training by less than 5%**, which is consistent with the cost in sCM and MeanFlow. The null token probability is 10% during pre-training, and the condition is never dropped during distillation.
>
> You are right that we omitted $t_{low}$: it is a hyperparameter ($0.05$ in our experiments) below which we do not use CFG for the teacher's instantaneous velocity prediction to improve stability in the high-noise region. We will add this to the revised paper.
>
> **[Q3] Cosine Similarity Loss Explanation**:
>
> **The cosine similarity loss is not a new contribution but a specific choice for aligning with our pre-trained teacher models (e.g., REPA, LightningDiT).** These models use a VAE whose features are trained with representation supervision (e.g., ssl model DINOv2). Our experiments showed that continuing to use this cosine term during DiT training beneficially constrains the feature space, and removing it increases FID by 0.1–0.2.
>
> **[Q4] Loss Weighting Analysis**:
>
> To maximize simplicity and avoid introducing excessive hyperparameters, our default approach is a direct summation of the two losses ($\lambda_{\text{FM}}=1.0$). However, you are right that the relative weighting is important, and our investigation reveals a nuanced picture that strongly supports this simple choice. The FM loss is a prerequisite for stability, but the minimum required weight depends on the model's initialization.
>
> 1.  For a model not pre-finetuned on our expanded time interval, a larger weight is needed for initial stability. As shown below, a weight of at least $\lambda_{\text{FM}} \geq 0.1$ is required to prevent collapse.
>
> | **FM Loss Weight ($\lambda_{\text{FM}}$)** | **FID (NFE=1, ↓)** |
> |:---:|:---:|
> | 0.0–0.1 | Collapse |
> | **0.1–10.0** | **3.17–3.22** |
> | 10.0–64.0 | 3.32–4.97 |
>
>
> 2.  In contrast, if the model is already finetuned with an FM objective, it is stable with even a minimal weight (e.g., $\lambda_{\text{FM}} \geq 10^{-8}$).
>
> | **FM Loss Weight ($\lambda_{\text{FM}}$)** | **FID (NFE=1, ↓)** |
> |:---:|:---:|
> | 0.0-1e-8 | Collapse |
> | 1e-8–1e-4 | 2.90-5.88 |
> | **1e-4–10.0** | **2.90–3.02** |
> | 10.0–64.0 | 3.02–4.58 |
>
> These results lead to a key conclusion: while a non-zero FM weight is essential, FACM is highly robust to the specific weight across several orders of magnitude once stability is achieved. **This robustness, which stems from our decoupled design, makes a direct summation (a weight of 1.0) a simple, effective, and reliable choice that avoids costly hyperparameter tuning.**

---

> > ### Comment · Reviewer_8cpy · 2025-11-26
> >
> > I thank the authors for answering my concerns. I have an additional question regarding the ablation table in the reply to reviewers AryK and 9H8q. Are those results obtained with or without CFG? And what architecture and architecture size are you using?

---

> > > ### Author Response · Authors · 2025-11-26
> > >
> > > **Response to Additional Question: Ablation Details (CFG and Architecture)**
> > >
> > > We thank the reviewer for this question regarding reproducibility and experimental setup. We are happy to clarify the details for the ablation results presented in our replies to Reviewer AryK and 9H8q:
> > >
> > > **1. Classifier-Free Guidance (CFG)**
> > >
> > > All FID results reported in the ablation tables (e.g., Table 3 and Table 4) are obtained **without Classifier-Free Guidance (CFG)**.
> > >
> > > * **Rationale:** The consistency model is trained through distillation. Our teacher model is explicitly trained to provide CFG, and the student (FACM) is trained to directly fit the trajectory of the teacher *with* CFG guidance applied. This means FACM naturally incorporates the effect of guidance during training, allowing for high-fidelity generation in a single NFE without requiring explicit CFG at inference time.
> > >
> > > **2. Architecture and Scale**
> > >
> > > The architecture and scale used for all ablation experiments are consistent with standard practice in this field:
> > >
> > > | Detail | Specification |
> > > | :--- | :--- |
> > > | **Architecture** | **LightningDiT-XL/2** [1] |
> > > | **Parameter Count** | **675 Million** parameters |
> > >
> > > The use of the established LightningDiT-XL/2 architecture ensures that our improvements are derived from the FACM method itself, rather than from novel architectural advances.
> > >
> > > [1] Reconstruction vs. Generation: Taming Optimization Dilemma in Latent Diffusion Models, CVPR 2025.

---

> ### Comment · Reviewer_8cpy · 2025-11-27
>
> I thank the authors for the clarification. I decided to raise my score.

---

> > ### Author Response · Authors · 2025-11-27
> > **Response to Reviewer 8cpy**
> >
> > We appreciate your constructive feedback and are grateful for the score increase.

---

### Official Review · Reviewer_9H8q · 2025-10-29

**Soundness:** 3
**Presentation:** 3
**Contribution:** 3
**Rating:** 6
**Confidence:** 3

**Summary:**

This paper identifies the cause of training instability problem for continuous-time consistency models and proposes a solution of Flow-Anchored Consistency Models(FACM) to solve it. The key design is the Flow-Anchoring principle, where a Flow Matching objective is used as a stable anchor and enables the model to learn both the flow's velocity field and the shortcut efficiently. Furthermore a memory-efficient Chain-JVP scheme is proposed which enables scalable FACM training up to a 14B parameter model. Experiment shows that the proposed algorithm achieves state-of-the-art results for few-step image generation task on a set of benchmarks including ImageNet 256x256 and CIFAR-10.

**Strengths:**

1. The analysis on the training instability issue in continuous-time consistency models is clear and convincing
2. The proposed flow-anchoring solution is elegant and effective, and the process of dual-objective training is clearly presented.
3. The achieved experiment result is impressive, where in table 1, FACM@NFE=2 outperforms Multi-NFE baseline of LightningDiT.

**Weaknesses:**

1. Although the paper shows efficiency in terms of reduced NFE, it would be good to also report the total generation time per image.
2. The individual components of the proposed algorithm including the flow matching loss and consistency model loss are existing work, and the proposed FSDP compatible Chain-JVP scheme still has significant computational overhead

**Questions:**

1. What is the computational overhead of the proposed Chain-JVP method?
2. What is the computational cost of distilling a 14B model and what is the GPU resources required?

---

> ### Author Response · Authors · 2025-11-21
> **Response to Reviewer 9H8q**
>
> **[W1] Generation Time**:
>
> Thank you for this practical question. While NFE is a critical metric, wall-clock time is also crucial for user experience. To provide a concrete comparison, we benchmarked the end-to-end generation time (including VAE decoding) per image on a single A100 GPU. Our 2-step FACM sampler takes approximately **70.2 ms** per image, whereas a standard 250-step Euler sampler takes **7062.9 ms**. This demonstrates that FACM is about **100 times faster** in practice, confirming that the significant reduction in NFE translates directly to a massive speed-up in wall-clock generation time.
>
> **[W2] Novelty of Components**:
>
> You are correct that CM and FM are existing concepts. **Our core novelty lies in how we combine them via a unified yet decoupled target.** We identified that prior CM approaches are unstable because they sacrifice endpoint quality (degenerating to noise) to satisfy global consistency. Our design directly addresses this: by anchoring the optimization to a predefined data flow and smoothly transitioning from CM to FM supervision at the endpoint, we enforce endpoint fidelity as a prerequisite for consistency. This principled design, which creates a unified target for a 'duplex' predictor, simultaneously ensures the stability and high fidelity of the final few-step model. This is a unique contribution that enables high-fidelity compression of complex teacher behaviors (e.g., from RLHF), which prior works could not achieve. Our results of detailed components ablation are summarized below:
>
> \begin{array}{lcc}
> \hline
> \text{Configuration} & \text{FID@Epochs 10 (NFE=1, } \downarrow \text{)} & \text{Collapse} \newline
> \hline
> \text{Fixed } r=1 \text{ MeanFlow (0\\% FM)} & \text{372.3} & \text{Yes} \newline
> \text{Fixed } r=1 \text{ MeanFlow (75\\% FM)} & \text{15.54} & \text{No} \newline
> \text{w/ Flow Anchoring (Expanded Time Interval)} & \text{4.31} & \text{No} \newline
> \text{w/ Interpolation} & \text{3.42} & \text{No} \newline
> \text{w/ Residual Clamping} & \text{2.86} & \text{No} \newline
> \text{w/ Beta Weighting} & \text{2.51} & \text{No} \newline
> \hline
> \end{array}
>
> These results confirm that our novel designs ensure the overall learning dynamic is stably guided by the FM anchor, leading to significantly better trajectory fidelity.
>
> **[Q1] Overhead and Cost of Chain-JVP**:
>
> The peak memory of Chain-JVP is identical to prior works (sCM with FlashAttention2), determined by one forward pass using Forward-mode Automatic Differentiation (Approximately equal to the memory cost of one backward pass). The main benefit of Chain-JVP is making CM training feasible for very large models under FSDP by **ensuring peak memory scales with the largest module, not the full model.** For a 14B model on 80GB A100s, Chain-JVP reduces peak memory from an **OOM** error to **~72GB**. For a 5B model, Chain-JVP and FA2 reduces memory from **~76G** to **~38G**. **Speed-wise, it is embedded in the FSDP backend and adds no overhead.**
>
> **[Q2] Computational Cost of 14B Model Distillation**:
>
> For the 14B model distillation, we used 64 A100 GPUs. The NFE=8 results shown in the paper were achieved after 5000 steps with a batch size of 512, which took 73 hours. This is reproducible on fewer GPUs (e.g., 8) with gradient accumulation and FSDP's CPU Offload.

---

### Official Review · Reviewer_YH7m · 2025-10-31

**Soundness:** 3
**Presentation:** 3
**Contribution:** 3
**Rating:** 6
**Confidence:** 4

**Summary:**

This paper proposes Flow-Anchored Consistency Models (FACM), a new framework for training consistency models that explicitly anchors the learned consistency mapping to an underlying flow-matching objective. The authors argue that existing continuous-time consistency models are unstable because they discard the flow-field information during training, leading to mode collapse or divergence. FACM mitigates this by jointly optimizing a flow-matching loss (to learn the velocity field) and a consistency loss (to learn efficient one-step mapping), decoupled through an expanded time interval strategy that ensures smooth continuity between the two objectives. The paper further introduces Chain-JVP, a memory-efficient Jacobian–vector product computation compatible with large-scale distributed training (e.g., FSDP), enabling the training of models up to 14B parameters. Experiments on CIFAR-10 and ImageNet-256 demonstrate that FACM achieves state-of-the-art one- and few-step generation performance, outperforming MeanFlow and sCM, and scales effectively to large text-to-image models.

**Strengths:**

1. The paper identifies a clear and well-motivated problem — instability in continuous-time consistency model training — and provides a principled remedy via flow anchoring.
2. The formulation is elegant, combining the benefits of flow matching (stability, theoretical grounding) and consistency models (few-step generation) into a unified loss.
3. The expanded time interval trick is simple yet effective, ensuring a smooth transition between FM and CM regions while avoiding gradient coupling issues.
4. The Chain-JVP technique is a practical and non-trivial engineering contribution, making second-order consistency training feasible at scale.
5. Empirical results are strong: FACM outperforms prior methods such as MeanFlow, sCM, and IMM in one- and two-step generation across multiple datasets.
6. The framework is demonstrated on a 14B-parameter text-to-image model, providing credible evidence of scalability beyond toy datasets.

**Weaknesses:**

1. The theoretical discussion remains somewhat heuristic. While the “anchoring” intuition is appealing, a more rigorous analysis (e.g., convergence guarantees or stability proofs) would strengthen the contribution.
2. The relationship between FACM and existing flow–consistency hybrids (e.g., MeanFlow, iCM) could be clarified — in particular, what distinguishes the proposed anchoring mechanism from joint FM/CM training used before.
3. Some ablations (e.g., varying the relative weight of FM vs. CM losses) are discussed briefly but lack quantitative depth.
4. The paper does not compare computational costs (training time, FLOPs) against prior CM methods, leaving unclear whether stability comes at significant overhead.

**Questions:**

1. How sensitive is FACM to the weighting between FM and CM objectives, and does this affect sample quality or convergence speed?
2. Could the authors clarify the precise role of the extended time interval—would a shared time embedding with gating suffice?
3. Does the Chain-JVP approach introduce any bias or approximation error compared to full JVP computation?

---

> ### Author Response · Authors · 2025-11-21
> **Response to Reviewer YH7m**
>
> **[W1] Rigorous Analysis of Flow-Anchoring**:
>
> We added a more rigorous analysis to the revised paper. **In fact, our Flow-Anchoring principle aims not only for the training stability of continuous-time consistency models but also for faster convergence and higher fidelity.** If only the CM task is trained, the model sacrifices endpoint quality after a gradient norm spike to satisfy global consistency. Pulled by this un-anchored consistency gradient, the model 'forgets' its flow memory of the velocity field $v$ and abandons the boundary condition. While introducing an FM objective can stabilize training, it still suffers from slow convergence and underfitting. **To further enforce endpoint fidelity as a prerequisite for global consistency,** we construct a unified yet decoupled target via an expanded time interval, and then diminish the CM gradient as inputs approach the data distribution, smoothly transitioning CM supervision into the FM supervision at the endpoint ($t=1$). This ensures the average velocity field is firmly anchored to the pre-defined flow.
>
> **[W3, Q1] Loss Weighting Analysis**:
>
> To maximize simplicity and avoid introducing excessive hyperparameters, our default approach is a direct summation of the two losses. Our investigation reveals a nuanced picture that strongly supports this simple choice.
>
> For a model not pre-finetuned on our expanded time interval, a larger weight is needed for initial stability. As shown below, a weight of at least $\lambda_{\text{FM}} \geq 0.1$ is required to prevent collapse.
>
> | **FM Loss Weight ($\lambda_{\text{FM}}$)** | **FID (NFE=1, ↓)** |
> |:---:|:---:|
> | 0.0–0.1 | Collapse |
> | **0.1–10.0** | **3.17–3.22** |
> | 10.0–64.0 | 3.32–4.97 |
>
>
> In contrast, if the model is already finetuned with an FM objective, it is stable with even a minimal weight (e.g., $\lambda_{\text{FM}} \geq 10^{-8}$).
>
> | **FM Loss Weight ($\lambda_{\text{FM}}$)** | **FID (NFE=1, ↓)** |
> |:---:|:---:|
> | 0.0-1e-8 | Collapse |
> | 1e-8–1e-4 | 2.90-5.88 |
> | **1e-4–10.0** | **2.90–3.02** |
> | 10.0–64.0 | 3.02–4.58 |
>
> These results lead to a key conclusion: while a non-zero FM weight is essential, **FACM is highly robust to the specific weight across several orders of magnitude once stability is achieved. This robustness, which stems from our decoupled design, makes a direct summation a simple, effective, and reliable choice that avoids costly hyperparameter tuning.**
>
> **[W2] Comparison with MeanFlow and iCM**:
>
> FACM is distinct from MeanFlow in its fundamental goal and design. Our primary goal is to achieve training for continuous-time CMs that is not only stable but also high-fidelity. In contrast, MeanFlow's standard approach, which generalizes the task to any time pair $(t, r)$, introduces several issues:
>
> 1.  **Uncertain Convergence:** Its endpoint $\mathbf{x}_r$ varies with $r$, making the final convergence distribution uncertain, which harms fidelity.
> 2.  **Inefficient and Conflicting Objectives:** Its CM and FM training objectives are coupled, causing them to interfere with each other's convergence.
> 3.  **Architectural Incompatibility:** Its dual-time conditioning requires architectural changes, which disrupts plug-and-play compatibility with existing models.
>
> Regarding iCM, its consistency optimization is based on a discrete-time formulation, which is prone to cumulative discretization errors that our continuous-time approach avoids.
>
> **[W4] Computational Costs**:
>
> The CM portion of FACM has an identical memory footprint and training speed to sCM and MeanFlow. MeanFlow optimizes the FM task in separate steps, requiring extra sampling and forward passes. In FACM, the FM target is efficiently obtained from a term already computed for the CM target. Therefore, per step, FACM has an identical memory requirement to sCM/MeanFlow, and it has two fewer non-gradient forward passes than MeanFlow. Crucially, FACM converges significantly faster, **we achieve the final FID of sCM and MeanFlow in just 1k steps of training.** Thus, the overall training cost of FACM is substantially lower than prior CM methods.
>
> **[Q2] Dual-Conditioning Design**:
>
> This is an excellent question. We initially explored simpler conditioning schemes like dual embeddings and gated conditions. **However, we found that these ad-hoc, non-continuous approaches all failed to converge.** This empirical result demonstrated that a mathematically principled formulation like our Expanded Time Interval—which ensures a smooth, continuous transition between the two task domains—is necessary to successfully multiplex the dual objectives.
>
> **[Q3] Chain-JVP vs. Full JVP**:
>
> **Chain-JVP is mathematically equivalent to a full JVP, as it introduces no bias or approximation error.** It is an engineering contribution that reorders the computation to be compatible with FSDP, ensuring the peak memory depends on the largest single module, not the entire model. We observed no reduction in computation speed compared to a standard FSDP forward pass.

---

### Official Review · Reviewer_AryK · 2025-11-04

**Soundness:** 3
**Presentation:** 3
**Contribution:** 3
**Rating:** 6
**Confidence:** 4

**Summary:**

This paper proposes Flow-Anchored Consistency Models (FACM), which incorporates a flow matching objective into consistency model training to stabilize training dynamics. The authors demonstrate that the key to effective joint optimization of flow matching and consistency model training is decoupling of flow velocity prediction and mean velocity prediction.

Specifically, in contrast to MeanFlow and its variants which are trained to predict flow velocity at time $t$ when given time condition $(t,t)$ and mean velocity from time $t$ to $r$ when given time condition $(t,r)$, FACMs are trained to predict flow velocity at $t$ given $2-t$ and mean velocity from time $t$ to $1$ given $t$. In contrast to MeanFlow, whose time conditions are coupled, i.e., $r = t$ for flow velocity, FACM uses decoupled time conditions, i.e., $r \neq t$ for flow velocity. Additional techniques such as interpolation of the shortcut target with the current EMA network output and scalable chain-JVP implementation are provided to further accelerate training.

The authors verify the scalability of FACM on CIFAR-10, ImageNet 256x256, and a Text-to-Image dataset. FACM is shown to consistently out-perform several fast baselines such as IMM, MeanFlow, and sCM.

**Strengths:**

- **[S1] This paper is original in the aspect that it provides a new perspective into training instability of CMs.** Specifically, the authors hypothesize that CM training instability arises from missing flow velocity supervision, and that one should decouple flow and mean velocity time conditions to mitigate conflict between the two velocities.

- **[S2] This paper is significant in the aspect that it provides a number of techniques for scaling CMs.** The authors provide a number of practical techniques, such as interpolation of shortcut target and current prediction and chain-JVP, which are shown to work on difficult tasks such as ImageNet 256x256 and text-to-image generation.

**Weaknesses:**

- **[W1] The paper lacks theoretical novelty, in the sense that it is a special case of MeanFlow.** MeanFlow learns a flow map between all time pairs $(t,r)$ for $0 \leq t < r \leq 1$, along with a flow matching loss at $t = r$. FACM is a special instance of MeanFlow where a flow map is learned only for time pairs $(t,1)$ for $0 \leq t \leq 1$, also with flow matching loss at $t = r$, where $r = 2 - c\_{FM}$ if one uses expanded time interval proposed in Section 3.3.2. Under this perspective, FACM may be viewed just as a collection of techniques for improved training of MeanFlow.

- **[W2] The paper is missing key ablations of the proposed techniques.** Under the MeanFlow perspective written in [W1], key techniques proposed in this paper can be categorized into (1) fixing $r = 1$, i.e., training only with time pairs $(t,1)$, (2) expanded time interval, and (3) other extraneous techniques such as shortcut target interpolation, residual clamping, and tuning $\alpha(t)$,$\beta(t)$ in Eq. (11) and (13). However, authors only provide an ablation of (2), so with all techniques intertwined, it is difficult to judge what is the largest contributor to the final generative performance.

- **[W3] The key hypothesis for instability of CMs, claimed by the authors, is unsupported.** At lines 171-175, the authors write "However, without a stable anchor in the underlying flow, the model's output $F_\theta$ quickly begins to drift. ... the derivative term in the identity grows to dominate the ground-truth velocity $v$, effectively diluting its supervisory signal." However, there is no supporting theory or experiment to verify this claim.

**Questions:**

**[Q1] Can the authors provide ablations mentioned in [W2] starting from a MeanFlow baseline?** In particular, I am curious how generative performance changes if (a) one fixes $r = 1$. I expect we would obtain a stronger few-step model, as it is solving an easier task compared to MeanFlow, which learns flow maps for all pairs $(t,r)$.

**[Q2] Can the authors provide theoretical or experimental evidence of the hypothesis that instability in CM is caused by the lack of instantaneously velocity field supervision?** The authors could, for instance, plot the variance of the derivative term $dF_{\theta^{-}}/dt$ without and with flow matching loss.

**[Q3] What is the purpose the cosine similarity term in Eq. (9)?** I believe this loss is redundant, as a cosine similarity is already implicitly contained in the first flow matching objective. If this term plays a non-trivial role, then it should also be included in the ablations.

---

> ### Author Response · Authors · 2025-11-21
> **Response to Reviewer AryK**
>
> **[W1] Compare with MeanFlow (TCM)**:
> Although FACM’s raw CM target coincides with MeanFlow when $r=1$, their optimization approaches are fundamentally different. **Our primary goal is to achieve training for continuous-time CMs that is not only stable but also high-fidelity.** We identify that the mode collapse in continuous-time CMs occurs because the model sacrifices endpoint quality in pursuit of global consistency. Based on this finding, FACM first introduces Flow Anchoring to achieve basic stable training requirements. To further enforce endpoint fidelity as a prerequisite for global consistency, we construct a unified yet decoupled target via Expanded Time Interval, and then diminish the CM gradient as inputs approach the data distribution, smoothly transitioning CM supervision into the FM supervision at the endpoint ($t=1$). This ensures the average velocity field is firmly anchored to the pre-defined flow. In contrast, the endpoint of MeanFlow $\mathbf{x}_r$ varies with $r$. This makes the final convergence distribution uncertain, which can affect its fidelity. Under identical teachers and settings, FACM converges much faster and attains lower FIDs than MeanFlow, indicating that our design changes the optimization geometry rather than merely fixing $r$.
>
> **[W2, Q1] Ablation of Key Components**:
> Thank you for this crucial question. We have conducted the detailed ablation you suggested on ImageNet 256x256 to dissect the contribution of each component. Our results are summarized below:
>
> \begin{array}{lcc}
> \hline
> \text{Configuration} & \text{FID@Epochs 10 (NFE=1, } \downarrow \text{)} & \text{Collapse} \newline
> \hline
> \text{Fixed } r=1 \text{ MeanFlow (0\\% FM)} & \text{372.3} & \text{Yes} \newline
> \text{Fixed } r=1 \text{ MeanFlow (75\\% FM)} & \text{15.54} & \text{No} \newline
> \text{w/ Flow Anchoring (Expanded Time Interval)} & \text{4.31} & \text{No} \newline
> \text{w/ Interpolation} & \text{3.42} & \text{No} \newline
> \text{w/ Residual Clamping} & \text{2.86} & \text{No} \newline
> \text{w/ Beta Weighting} & \text{2.51} & \text{No} \newline
> \hline
> \end{array}
>
> These results confirm that Flow-Anchoring (the FM objective) is the fundamental prerequisite for stability. Furthermore, our Expanded Time Interval design decouples the FM and CM tasks into different time intervals. This allows FACM to achieve the fastest convergence, even when the CM loss formulation is identical to that of the baselines. The auxiliary techniques then work in synergy to achieve high fidelity. Specifically, shortcut interpolation ($\alpha$) and beta weighting ($\beta$) ensure a smooth transition from CM to FM supervision as $t \to 1$, preventing the model from sacrificing endpoint quality for global consistency. Residual clamping is an empirical technique that further stabilizes training by preventing gradient spikes from BF16 JVP computation. **Together, these components ensure the overall learning dynamic is stably guided by the FM anchor, leading to significantly better trajectory fidelity.**
>
> **[W3, Q2] Training Instability of CMs**:
> Thank you for this question. Our hypothesis originates from a key observation: when training a pure continuous-time CM, we invariably see the total gradient norm spike (a visualization is provided in Fig. 3), after which the model’s output degenerates into pure noise. **This indicates that the model sacrifices endpoint quality to satisfy the global consistency. In the optimization objective, the teacher-provided velocity $v$ is reliable, making the derivative term $d F_\theta / dt$ the only source of instability.** While the magnitude and variance of this term show no anomaly during the spike, it provides a substantial consistency gradient, causing the model to 'forget' its flow memory of $v$ itself and abandon the boundary condition. Our FACM strategy is explicitly designed to prevent this failure mode. By constructing a unified target where the CM gradient diminishes and smoothly transitions to the FM supervision as $t \to 1$, we enforce endpoint fidelity as a prerequisite for achieving global consistency. This anchors the optimization to the **correct data manifold**, ensuring stable training. **The updated experiment in Table 5** further supports our CM Instability hypothesis.
>
> **[Q3] Cosine Similarity Term**:
> Thank you for this question. The cosine similarity term is not redundant. As part of our ablation studies, we confirmed its contribution. It is included specifically for alignment with our pre-trained model, LightningDiT, which uses a VA-VAE trained with representation supervision. Including the cosine similarity term during DiT training constrains its features in a beneficial way. Removing this loss increases the FID by 0.1–0.2, confirming its non-trivial role in achieving our final performance.

---

> ### Comment · Reviewer_AryK · 2025-11-25
>
> Thank you for the detailed reply. I have additional questions, and hope the authors can clarify them.
>
> **Regarding the authors' reply to [W1] -- lack of theoretical novelty**
>
> I have difficulty seeing how the authors' reply to [W1] adds non-trivial theoretical insight into the current manuscript. While the authors claim
>
> > Under identical teachers and settings, FACM converges much faster and attains lower FIDs than MeanFlow, indicating that our design changes the optimization geometry rather than merely fixing $r$
>
> in the rebuttal, the authors do not provide any theoretical derivation showing that FACM, for instance, reduce gradient variance. I feel that any derivation, even under a toy setup, would make this paper really solid. One could look at Proposition 1 of [1] for an example of a good theoretical analysis which provides useful insights into optimization dynamics of CMs.
>
> [1] Improved Techniques for Training Consistency Models, ICLR, 2024.
>
> **Regarding the authors' reply to [W2,Q1] -- missing ablations**
>
> The provided ablation is really helpful, and clarifies the contribution of each technique. However, I believe FID for MeanFlow with 75% FM (without fixed $r$) is missing from the table.
>
> **Regarding the authors' reply to [Q3] -- cosine similarity term**
>
> In the rebuttal, the authors write
>
> > The cosine similarity term is included specifically for alignment with our pre-trained model, LightningDiT, which uses a VA-VAE trained with representation supervision. Including the cosine similarity term during DiT training constrains its features in a beneficial way.
>
> What do the authors mean by "constraining features in a beneficial way"? Is there an auxiliary representation alignment loss on the consistency model not mentioned in the paper?

---

> > ### Author Response · Authors · 2025-11-26
> > **Response to Reviewer AryK**
> >
> > **Response to [W1] -- Lack of Theoretical Novelty**
> >
> > We share the reviewer's view on the critical importance of theoretical rigor. In response, we have added **Appendix A.1 (highlighted in red)**, which formally derives FACM's stability and convergence properties:
> >
> > 1.  **Gradient Norm Analysis (Appendix A.1.1):** We decompose the CM gradient and prove that pure CM suffers from gradient explosion because **both the parameter sensitivity and the prediction error are unbounded**. Even when using a Normed L2 loss, this unboundedness manifests as a catastrophic gradient spike. We show that FACM's Flow Matching branch acts as a spectral regularizer, imposing a global Lipschitz bound that prevents this divergence.
> >
> > 2.  **Proof of Variance Reduction (Appendix A.1.2, Step 2.1):** We rigorously prove that our weighting schedule $\alpha(t)$ asymptotically cancels the $(1-t)^{-1}$ singularity in the CM target. Using a Taylor expansion around $t=1$, we demonstrate that this mechanism **significantly limits the target variance**, addressing the instability caused by the singularity.
> >
> > 3.  **Convergence Guarantee (Appendix A.1.2, Step 2.2):** We derive a finite bound for the FACM gradient variance. Unlike pure CM's "moving target" problem, we show that FACM's gradient variance is strictly controlled by the network's **Lipschitz constant** (inherited via the shared parameters) and the **variance-reduced target**, ensuring stable convergence.
> >
> > ***
> >
> > **Response to [W2, Q1] -- Missing MeanFlow Ablations**
> >
> > We have added the requested experiment for MeanFlow (Variable $r$) with 75% FM in Table 4, while the **final convergence comparison between MeanFlow (75% FM) and FACM** already appears in the **original Table 3** of the manuscript
> >
> > | Configuration | FID (NFE=1, $\downarrow$) | Collapse |
> > | :--- | :---: | :---: |
> > | MeanFlow (Variable $r$, 75% FM) | 43.03 | No |
> > | MeanFlow (Fixed $r=1$, 75% FM) | **15.54** | **No** |
> >
> > While adding 75% FM to the standard MeanFlow prevents collapse, it converges significantly slower (FID 43.03) compared to fixing $r=1$ (FID 15.54).
> >
> > ***
> >
> > **Response to [Q3] -- Clarification on Cosine Similarity**
> >
> > **The cosine similarity loss effectively serves as a representation alignment loss**. Specifically, the teacher model (VA-VAE [1]) utilizes both MSE and cosine similarity losses to supervise its Flow Matching task. Consequently, FACM incorporates this cosine loss to strictly align with the teacher's native optimization landscape and feature geometry, without the need for any additional auxiliary losses.
> >
> > [1] Reconstruction vs. Generation: Taming Optimization Dilemma in Latent Diffusion Models, CVPR 2025.

---

### Author Response · Authors · 2025-11-21
**Summary of Revisions for All Reviewers**

We thank all reviewers for their constructive feedback and have incorporated their suggestions into the revised paper. Key revisions include:

*   **New Theoretical Analysis:** Added a complete theoretical derivation in **Appendix A.1**, including (1) gradient norm decomposition and stability analysis, (2) formal proof of variance reduction through the α(t) schedule, and (3) convergence guarantees showing bounded gradient variance due to shared-parameter Lipschitz constraints. All subsequent appendices have been renumbered accordingly (e.g., former Appendix A.7 → A.8, A.8 → A.9, etc.).

*   **New Ablation & Sensitivity Studies:** Added component ablations (Table 4), loss weighting sensitivity (Table 5), and gradient norm analysis (Figure 3a) in Section 4.3.

*   **Implementation Clarifications:** Added details on cosine similarity loss (Appendix A.8) and CFG strategy (Appendix A.4d).

*   **Computational Benchmarks:** Added Appendix A.9 reporting wall-clock latency, peak memory, and 14B training resources.

In addition to these reviewer-specific revisions, we have further expanded the **Additional Visualization section (Appendix A.10)** with uncurated text-to-image (T2I) results from our 14B model across NFE = 2, 4, 6, and 8 to provide a more comprehensive qualitative comparison.

---

### Meta-Review · Area_Chair_DGLH · 2025-12-17

**Summary:**

This paper tackles an important problem: the instability and scalability of continuous-time, consistency-style training for few-step generative models. It presents a coherent framework linking flow matching, shortcut learning, and large-scale distillation, supported by strong experiments on CIFAR-10, ImageNet, and a 14B-parameter text-to-image model. The ability to scale exact JVP-based training under FSDP is a meaningful systems contribution. The method is simple, architecture-agnostic, and well validated through thorough ablations, and the results convincingly show that instantaneous-velocity supervision can stabilize derivative-based shortcut training at scale.

Overall, this is a solid and technically competent paper with strong empirical support and a meaningful contribution. Its impact would be improved by clearer positioning within the consistency model literature, more explicit attribution to the MeanFlow framework, and a more precise statement of the scope under which flow anchoring is necessary. With these clarifications addressed in revision, considering the outcome of the reviews and discussion, I support acceptance.

**Reviewer Concerns:**

There are notable concerns about framing and positioning. The paper introduces consistency models almost exclusively through a continuous-time derivative constraint, with little discussion of standard discrete-time or endpoint-based formulations. As written, a reader unfamiliar with the literature could reasonably infer that MeanFlow-style, JVP-based training is the canonical form of consistency models. This matters because several core claims,  particularly the instability diagnosis and the need for a “flow anchor”,  apply specifically to this MeanFlow-like formulation rather than to consistency models in general, where anchoring typically comes from boundary conditions and endpoint supervision. The exposition also closely follows known MeanFlow and flow-mapping derivations, and while prior work is cited, the presentation initially gives the impression of greater novelty than is warranted. The main contribution lies less in the combination of losses itself, which has precedent, and more in the decoupling of shortcut and flow objectives via conditioning and in the scalable JVP implementation.

**Reviewer Scores:**

I think that the score would have stay approximately at the current level, with possible slight improvements, due to the novelty/positioning concerns.

---

### Decision · Program_Chairs · 2026-01-26

Accept (Poster)